# On the (Non-)Robustness of Two-Layer Neural Networks in Different Learning Regimes

## ABSTRACT

Neural networks are known to be highly sensitive to adversarial examples. These may arise due to different factors, such as random initialization, or spurious correlations in the learning problem. To better understand these factors, we provide a precise study of the adversarial robustness in different scenarios, from initialization to the end of training in different regimes, as well as intermediate scenarios, where initialization still plays a role due to "lazy" training. We consider over-parameterized networks in high dimensions with quadratic targets and infinite samples. Our analysis allows us to identify new tradeoffs between approximation (as measured via test error) and robustness, whereby robustness can only get worse when test error improves, and vice versa. We also show how linearized lazy training regimes can worsen robustness, due to improperly scaled random initialization. Our theoretical results are illustrated with numerical experiments.

## 1 INTRODUCTION

Deep neural networks have enjoyed tremendous practical success in many applications involving high-dimensional data, such as images. Yet, such models are highly sensitive to small perturbations known as adversarial examples (Szegedy et al., 2013), which are often imperceptible by humans. While various strategies such as adversarial training (Madry et al., 2018) can mitigate this vulnerability empirically, the situation remains highly problematic for many safety-critical applications like autonomous vehicles and health, and motivates a better theoretical understanding of what mechanisms may be causing this.

Various factors are known to contribute to adversarial examples. In linear models, features that are only weakly correlated with the label, possibly in a spurious manner, may improve prediction accuracy but induce large sensitivity to adversarial perturbations (Tsipras et al., 2019; Sanyal et al., 2021). On the other hand, common neural networks may exhibit high sensitivity to adversarial perturbations at random initialization (Simon-Gabriel et al., 2019; Daniely & Shacham, 2020; Bubeck et al., 2021). While such settings already capture interesting phenomena behind adversarial examples, they are restricted to either trained linear models, or nonlinear networks at initialization. Trained, nonlinear networks may thus involve multiple sources of vulnerability arising from initialization, training algorithms, as well as the data distribution. Capturing the interaction between these different components is of crucial importance for a more complete understanding of adversarial robustness.

In this paper, we study the interplay between these different factors by analyzing approximation (i.e how well the model fits the data) and robustness properties (i.e the sensitivity of the model's outputs w.r.t perturbations in test data) of two-layer neural networks in different learning regimes. We consider two-layer finite-width networks in high dimensions with infinite training data, in asymptotic regimes inspired by Ghorbani et al. (2019). This allows us to focus on the effects inherent to the data distribution and the inductive bias of architecture (choice of activation function, number of hidden neurons per input dimension, etc.) and training algorithms, while side-stepping issues due to finite samples. Following Ghorbani et al. (2019), we focus on nonlinear regression settings with structured quadratic target functions, and consider commonly studied training regimes for two-layer networks, namely (i) neural networks with quadratic activations trained with stochastic gradient descent on the population risk which finds the global optimum; (ii) random features (RF, Rahimi & Recht, 2008), (iii) neural tangent (NT, Jacot et al., 2018), as well as (iv) "lazy" training (Chizat et al., 2019) regimes for basic RF and NT, where we consider a first-order Taylor expansion of the

network around initialization, including the initialization term itself (in contrast to the standard RF and NT regimes which ignore the offset due to initialization). Note that, though the theoretical setting is inspired by Ghorbani et al. (2019), our work differs from theirs in its focus and scope. Indeed, we are concerned with robustness and its interplay with approximation, in different learning regimes, while they are only concerned with approximation. We also note that the lazy/linearized regimes we study as part of this work were not considered by Ghorbani et al. (2019), and help us highlight the impact of initialization on robustness.

**Note that unlike the other regimes, SGD exhibits a kind of feature learning, whereby the first layer weights are learning specific directions by approximating the matrix $B$. In particular, this involves non-trivial feature selection via non-linear learning, while the other regimes (RF and NT) are linear estimators on top of non-linear but fixed features.**

**Main contributions.**   Our work establishes theoretical results which uncover novel tradeoffs between approximation (as measured via test error) and robustness that are inherent to all the regimes considered. These tradeoffs appear to be due to misalignment between the target function and the input distribution (weight distribution) for random features (Section 4), or to the inductive bias of fully-trained networks (Section 3 and Appendix E). We also show that improperly scaled random initialization can further degrade robustness in lazy/linearized models (Section 5), since the resulting models might inherit the nonrobustness inherent to random initialization. This raises the question of how small should the initialization be in order to enhance the robustness of the trained model. Our theoretical results are empirically verified with extensive numerical experiments on simulated data.

The setting of our work is regression in a student-teacher setup where the student is a two-layer feedforward neural network and the teacher is a quadratic form. We assume access to infinite training data. Thus, the only complexity parameters are the coefficient matrix of the teacher model, the input dimension $d$ and the width of the neural network $m$, assumed to both "large" but proportional to one another. Refer to Section 2 for details. In Appendix I, we also show similar but weaker trade-offs for arbitrary student and teacher models. The infinite-sample setting allows us to focus on the effects inherent to the data distribution and the inductive bias of architecture (choice of activation function) and different learning regimes, while side-stepping issues due to finite samples and label noise. Also note that in this infinite-data setting, label noise provably has no influence on the learned model, in all the learning regimes considered. The observation that there is a tradeoff between robustness and approximation, even in this infinite-sample setting, is one of the surprising findings of our work. This complements related works such as (Bubeck et al., 2020b; Bubeck & Sellke, 2021), which show that finite training samples with label noise is a possible source of nonrobustness in neural networks.

**Related works.**   Various works have theoretically studied adversarial examples and robustness in supervised learning, and the relationship to ordinary test error / accuracy. Tsipras et al. (2019) considers a specific data distribution where good test error implies poor robustness. Shafahi et al. (2018); Mahloujifar et al. (2018); Gilmer et al. (2018); Dohmatob (2019) show that for high-dimensional data distributions which have concentration property (e.g., multivariate Gaussians, distributions satisfying log-Sobolev inequalities, etc.), an imperfect classifier will admit adversarial examples. On the other hand, Yang et al. (2020) observed empirically that natural images are well-separated, and so locally-lipschitz classifies shouldn't suffer any kind of test error vs robustness tradeoff. However, gradient-descent is not likely to find such models. Our work studies regression problems with quadratic targets, and shows that there are indeed tradeoffs between test error and robustness which are controlled by the learning algorithm / regime and model. Schmidt et al. (2018); Khim & Loh (2018); Yin et al. (2019); Bhattacharjee et al. (2021); Min et al. (2021b;a) study the sample complexity of robust learning. In contrasts, our work focuses on the case of infinite data, so that the only complexity parameters are the input dimension $d$ and the network **width** $m$.

Gao et al. (2019); Bubeck et al. (2020b); Bubeck & Sellke (2021) show that over-parameterization may be necessary for robust interpolation in the presence of noise. In contrast, our paper considers a structured problem with noiseless signal and infinite training data, where the network width $m$ and the input dimension $d$ tend to infinity proportionately. In this under-complete asymptotic setting, our results show a systematic and precise tradeoff between approximation (test error) and robustness in different learning regimes. Thus, our work nuances the picture presented by previous works by exhibiting a nontrivial interplay between robustness and test error, which persists even in the case of infinite training data where the resulting model isn't affected by label noise.

**Dohmatob (2021); Hassani & Javanmard (2022) study the tradeoffs between interpolation, predictive performance (test error), and robustness for finite-width over-parameterized networks in kernel regimes with noisy linear target functions. In contrast, we consider structured quadratic target functions and compare different learning settings, including SGD optimization in a non-kernel regime, as well as lazy/linearized models.**

**Robustness has also been studied from a causality perspective. For example, Rothenhausler et al. (2021) studies tradeoffs between test error and robustness in linear regression under distributional shifts on the marginal distribution of the covariates.**

We provide a more detailed discussion of the related work in Appendix A.

*Remark 1.1. Note that the term "over-parametrization" is not used in our paper in the same sense as in Bubeck et al. (2020b); Bubeck & Sellke (2021); Hassani & Javanmard (2022). In those works, the setup is finite samples ($n < \infty$), and over-parametrization means that $m$ is substantially larger than $n/d$, where $d$ is the input-dimension, and $m$ is the network with (i.e number of neurons in the hidden layer). In our work, we focus on the infinite-sample case ($n = \infty$), and over-parametrization means $m/d$ is large. Finally, the findings of Bubeck et al. (2020b); Bubeck & Sellke (2021) –namely, that over-parametrization is beneficial for robustness, and of Hassani & Javanmard (2022) –namely, that over-parametrization is detrimental to robustness, are nuanced by Zhu et al. (2022) which shows that as the width $m$ of a neural network is increased, there transition from over-parametrization being detrimental, to being benificial for robustness. More precisely, they derived upper-bounds for robustness error which show that there critical value $m_0$ such that the robustness error is an increasing function of width in the interval $[1, m_0]$ (over-parametrization hurts) and a decreasing function of width in the interval $[m_0, \infty)$ (i.e over-parametrization becomes beneficial). Overall, the exact role of over-parametrization in robustness remains partly unclear, even though progress is being made on the subject.*

## 2 PRELIMINARIES

**Notations.** We use standard notations in our manuscript. A cheat sheet is provided in supplemental / appendix.

### 2.1 THE TEACHER MODEL: A QUADRATIC FORM

We consider the following regression setup proposed by Ghorbani et al. (2019). Let $B$ be a fixed $d \times d$ psd matrix and let $b_0 \in \mathbb{R}$ be a fixed unknown scalar. Consider the quadratic *teacher* model

$$f_\star(x) := x^\top B x + b_0, \text{ for any } x \in \mathbb{R}^d. \tag{1}$$

We assume the input data is distributed according to $N(0, I_d)$, the standard Gaussian distribution in $d$ dimensions. Thus, the structure of the problem of learning the teacher model $f_\star$ in (1) is completely determined by the unknown $d \times d$ matrix $B$. We assume an idealized scenario where the learner has access to an infinite number of iid samples of the form $(x, f_\star(x))$ width $x \sim P_x := N(0, I_d)$. For simplicity of analysis, we will further assume as in (Ghorbani et al., 2019) that the teacher model $f_\star$ defined in (1) is centered, i.e $\mathbb{E}_{x \sim N(0, I_d)}[f_\star(x)] = 0$. This forces the offset $b_0 = -\text{tr}(B)$.

### 2.2 THE STUDENT MODEL: A TWO-LAYER NEURAL NETWORK

Consider a two-layer *student* neural network

$$f_{W,z,s}(x) := \sum_{j=1}^m z_j \sigma(x^\top w_j) + s, \tag{2}$$

where $m$ is the network width, i.e., the number of hidden neurons each with parameter vector $w_j \in \mathbb{R}^d$, output weights $z = (z_1, \dots, z_m) \in \mathbb{R}^m$, and activation function $\sigma : \mathbb{R} \to \mathbb{R}$. We define $W$ as the $m \times d$ matrix with $j$th row $w_j$. The scalar $s$ is an offset which we will sometimes set to 0, in which case we will simply write $f_{W,z,s} := f_{W,z,0}$. Note that the teacher model $f_\star$ is itself a two-layer

neural network with output weights fixed to 1, quadratic activation function, and $m = d$ hidden neurons with parameters $W_\star := B^{1/2}$, where $B^{1/2}$ is the unique psd matrix such that $(B^{1/2})^2 = B$.

The aim of learning is to approximate the teacher model $f_\star$ as closely as possible with the student model $f_{W,z,s}$. We will consider the following high-dimensional setup:

– **Infinite training data**, wherein the sample size $n$ is *equal to* $\infty$, i.e., the learner has access to the entire data distribution, allowing us to step-aside issues linked with finite samples.

– **Proportionate scaling** of input dimension $d$ and student network width $m$, wherein $d$ and $m$ are *finite*, and large of the same order, i.e.,

$$m, d \to \infty, \ m/d \to \rho \in (0, \infty). \tag{3}$$

The *parameterization rate* $\rho$ (which corresponds to the number of hidden neurons per input dimension), will play a crucial role in our analysis. The case $\rho < 1$ corresponds to *under-parametrization*, while $\rho > 1$ corresponds to *over-parametrization*. Occasionally, we will also consider the extreme over-parametrization regime corresponding to $\rho \gg 1$, or more precisely, the limiting case $\rho \to \infty$.

### 2.3 METRICS FOR TEST ERROR AND ROBUSTNESS

**Test error.** The test / approximation error of a student model $f : \mathbb{R}^d \to \mathbb{R}$ is defined by

$$\varepsilon_{\text{test}}(f) := \|f - f_\star\|^2_{L^2(P_x)} = \mathbb{E}_{P_x}(f(x) - f_\star(x))^2, \tag{4}$$

where $P_x$ is the distribution of the features. (4) measures how well the student $f$ approximates the teacher model $f_\star$. *Except otherwise explicitly stated, in this article we will always consider the isotropic case where the distribution of the features is $P_x = N(0, I_d)$, as in Ghorbani et al. (2019).*

It will be instructive to compare the test error of $f$ to that of the *null predictor* which outputs 0 on every input, namely $\|f_\star\|^2_{L^2(P_x)}$. Thus, consider the *normalized test error*,

$$\widetilde{\varepsilon}_{\text{test}}(f) := \varepsilon_{\text{test}}(f)/\|f_\star\|^2_{L^2(P_x)}. \tag{5}$$

This quantity was studied in Ghorbani et al. (2019) where explicit analytic formulae were obtained for two-layer networks in various regimes of interest: networks fully trained by stochastic gradient-descent (SGD) on the population risk, random features (RF), and neural tangent (NT). We shall consider these same regimes and establish tradeoffs between test error and robustness of the corresponding models. This will paint a picture complementary to Ghorbani et al. (2019).

**Measure of robustness / sensitivity.** We will measure the robustness of a smooth student model $f : \mathbb{R}^d \to \mathbb{R}$ (e.g., the two-layer neural net (2)) by what we call its *robustness error*, defined as the square-root of its Dirichlet energy $\varepsilon_{\text{rob}}(f)$ w.r.t. to a random test point $x \sim P_x$, that is

$$\varepsilon_{\text{rob}}(f) := \|\nabla_x f\|^2_{L^2(P_x)} = \mathbb{E}_{P_x}\|\nabla_x f(x)\|^2. \tag{6}$$

Smoothness here is in the very general sense of Gigli & Ledoux (2013, Section 4.1) with euclidean structure. The smaller the value of $\varepsilon_{\text{rob}}(f)$, the more robust / less sensitive $f$ is to changes in a test data point, on average. We justify the choice of this quantity as a measure of robustness in Appendix D, where we will link it to more classical notions of robustness error (Madry et al., 2018). In particular, the teacher model has $\varepsilon_{\text{rob}}(f_\star) = 4\|B\|^2_F$. Finally, note that, measures of robustness based on notions of sensitivity have been considered in other works like Bubeck et al. (2020b); Bubeck & Sellke (2021) for regression, and Wu et al. (2021) for classification settings.

It will be convenient to compare the robustness of a student model $f$ to that of the baseline quadratic teacher model $f_\star$ defined in (1). To this end, consider the *normalized robustness error* $f$ defined by

$$\widetilde{\varepsilon}_{\text{rob}}(f) := \varepsilon_{\text{rob}}(f)/\varepsilon_{\text{rob}}(f_\star), \tag{7}$$

which measures the relative robustness error of the student. The objective of our paper is to study the quantity $\widetilde{\varepsilon}_{\text{rob}}(f)$ for neural networks (2) in various regimes in the limit (3), and put to light interesting phenomena. In particular, we will establish tradeoffs between test error and robustness error, in the form of a nontrivial relationship

$$\widetilde{\varepsilon}_{\text{test}}(f) + \widetilde{\varepsilon}_{\text{rob}}(f) = 1, \tag{8}$$

for different learning regimes. This paints a picture complementary to Ghorbani et al. (2019).

**Remark 2.1.** *Note that it might be tempting to thing that*

$$\|f - f_\star\|_{L^2(P_x)} \approx 0 \implies \|\nabla f\|_{L^2(P_x)} \approx \|\nabla f_\star\|_{L^2(P_x)}, \tag{9}$$

*Such an implication would automatically lead to a (at least) heuristic explanation of our tradeoffs (8). However, (9) is false in general. Indeed, a smooth function (small $\|\nabla f\|_{L^2(P_x)}$) can be approximated very well in $L^2$ by functions by very rough functions (large $\|\nabla f\|_{L^2(P_x)}$). This point is elaborated in Appendix K. To establish our tradeoffs, we exploit the fine structure of two-layer neural networks in the different learning regimes considered.*

## 3 RESULTS FOR TWO-LAYER NEURAL NETWORKS TRAINED VIA SGD

Consider a student neural network model with quadratic activation $f_{\mathrm{SGD}} : \mathbb{R}^d \to \mathbb{R}$, i.e

$$f_{\mathrm{SGD}}(x) := \sum_{j=1}^{m} (x^\top w_j)^2 + s. \tag{10}$$

Here, $W = (w_1, \ldots, w_m) \in \mathbb{R}^{m \times d}$ is a matrix of learnable parameters (one per hidden neuron), and $s \in \mathbb{R}$ is a learnable offset. The output weights vector is fixed to $z = 1_m := (1, \ldots, 1)$, while $W$ and $s$ are optimized via SGD (hence the subscript), where each update is on a single new sample point. It is shown in Theorem 3 of Ghorbani et al. (2019) that if $W_t \in \mathbb{R}^{m \times d}$ is the matrix of hidden parameters after $t$ steps of SGD, then in the limit (3), the matrix $W_t W_t^\top \in \mathbb{R}^{m \times m}$ converges a.s to the best rank-$m$ approximation of $B$. Thus, by continuity of matrix norms, we deduce that $\|W_t W_t^\top\|_F^2$ converges a.s to $\|B\|_{F,m}^2$, in the infinite data limit $t \to \infty$.

Combining with Lemma G.1 establishes the following asymptotic formula for the (normalized) robustness of the resulting model $f_{\mathrm{SGD}}$, in the high-dimensional limit (3).

**Theorem 3.1.** *In the limit (3), it holds that* $\widetilde{\varepsilon}_{\mathrm{rob}}(f_{\mathrm{SGD}}) \xrightarrow{a.s} \dfrac{\|B\|_{F,m}^2}{\|B\|_F^2} = \dfrac{\sum_{k=1}^{m \wedge d} \lambda_k(B)^2}{\sum_{j=1}^{d} \lambda_k(B)^2} \leq 1$, *with*

*equality iff* $\mathrm{rank}(B) \leq m$. *In particular, if $\rho \geq 1$, then in the limit (3), it holds that* $\widetilde{\varepsilon}_{\mathrm{rob}}(f_{\mathrm{SGD}}) \xrightarrow{a.s} 1$.

**Tradeoff approximation and robustness.** We see from the above theorem that if $m \geq \mathrm{rank}(B)$, then the robustness error for the learned student converges to that of the true model if $m \geq d$, namely $\varepsilon_{\mathrm{rob}}(f_{\mathrm{SGD}}) \xrightarrow{p} \sum_{j=1}^{d} \lambda_j(B)^2 = \varepsilon_{\mathrm{rob}}(f_\star)$. This is the case if $m \geq d$, for example. Otherwise (i.e if $m < \mathrm{rank}(B)$), then the limiting value of $\varepsilon_{\mathrm{rob}}(f_{\mathrm{SGD}})$ can be arbitrarily less than $\varepsilon_{\mathrm{rob}}(f_\star)$, i.e., the learned student will be much more robust (i.e., stable) than the ground truth model. Comparing with Theorem 3 and Proposition 1 of Ghorbani et al. (2019), we can see that any decrease in robustness error of the learned student (compared to the teacher model) is at the expense of increased test error, and vise versa. Indeed, it was shown in that paper that the normalized test error $\widetilde{\varepsilon}_{\mathrm{test}}(f_{\mathrm{SGD}})$ verifies

$$\widetilde{\varepsilon}_{\mathrm{test}}(f_{\mathrm{SGD}}) \xrightarrow{p} 1 - \lim \|B\|_{F,m}^2 / \|B\|_F^2, \text{ in the limit (3)} \tag{11}$$

Combining with our Theorem 3.1 above, we deduce that

$$\widetilde{\varepsilon}_{\mathrm{test}}(f_{\mathrm{SGD}}) + \widetilde{\varepsilon}_{\mathrm{rob}}(f_{\mathrm{SGD}}) \xrightarrow{p} 1, \text{ in the limit (3).} \tag{12}$$

The above formula highlights a tradeoff between test error and robustness error. Thus, we have identified a novel tradeoff between approximation and robustness for the neural network model (2) trained via SGD. In the sequel, we shall establish such tradeoffs for other learning regimes.

## 4 RESULTS FOR THE RANDOM FEATURES MODEL

Consider the two-layer model (2) with hidden neuron parameters $w_1, \ldots, w_m$ sampled iid from a $d$-dimensional multivariate Gaussian distribution $N(0, \Gamma)$ with covariance matrix $\Gamma$. We denote this so-called random features (RF) student model $f_{\mathrm{RF}}$, defined by

$$f_{\mathrm{RF}}(x) = f_{W, z_{\mathrm{RF}}}(x) = z_{\mathrm{RF}}^\top \sigma(Wx), \tag{13}$$

where $z_{\mathrm{RF}} \in \mathbb{R}^m$ solves the following linear regression problem

$$\arg \min_{z \in \mathbb{R}^m} \mathbb{E}_{x \sim N(0, I_d)}[(z^\top \sigma(Wx) - f_\star(x))^2]. \tag{14}$$

It is easily seen that for $x \sim N(0, I_d)$, one has $z_{\mathrm{RF}} = U^{-1} v$, with

$$U_{jk} := \mathbb{E}_x[\sigma(x^\top w_j)\sigma(x^\top w_k)] \,\forall j, k \in [m], \tag{15}$$

$$v_j := \mathbb{E}_x[f_\star(x)\sigma(x^\top w_j)] \,\forall j \in [m]. \tag{16}$$

The covariance matrix $\Gamma$ encompasses the inductive bias of the neurons at initialization to different directions in feature space. Define the alignment $\alpha = \alpha(B, \Gamma)$ of the hidden neurons to the task at hand, namely learning the teacher model $f_\star$, as follows

$$\alpha := \frac{\mathrm{tr}(B\Gamma)}{\|B\|_F \|\Gamma\|_F} \le 1. \tag{17}$$

As we shall see, the task-alignment $\alpha$ plays a crucial role in the dynamics of prediction performance (test error) and robustness $f_{\mathrm{RF}}$.

## 4.1 Assumptions and key quantities

As in Ghorbani et al. (2019), we will need the following mild technical conditions on $\Gamma$.

**Condition 4.1.** *The covariance matrix $\Gamma$ satisfies: (A) $\mathrm{tr}(\Gamma) = 1$ and $d \cdot \|\Gamma\|_{op} = \mathcal{O}(1)$. (B) The empirical eigenvalue distribution of $d \cdot \Gamma$ converges weakly to a probability distribution $\mathcal{D}$ on $\mathbb{R}_+$.*

This condition is quite reasonable, and moreover, it allows us to leverage standard tools from random matrix theory (RMT) in our analysis. We will also need the following technical condition on the activation function $\sigma$.

**Condition 4.2.** *$\sigma$ is weakly continuously-differentiable and satisfies the growth condition $\sigma(t)^2 \le c_0 e^{c_1 t^2}$ for some $c_0 > 0$ and $c_1 < 1$, and for all $t \in \mathbb{R}$. Moreover, $\sigma$ is not a purely affine function.*

The above growth condition is a classical condition usually imposed for the theoretical analysis of neural networks (see, e.g., Ghorbani et al., 2019; Mei & Montanari, 2019; Montanari & Zhong, 2020), and is satisfied by all popular activation functions used in practice. One of its main purposes is to ensure that all the Hermite coefficients $(\lambda_k)_{k \in \mathbb{N}}$ of the activation function exist. Refer to Section G.2 for precise definition of Hermite coefficients. We will also need the following condition.

**Condition 4.3.** *(A) $\lambda_0 := \lambda_0(\sigma) = 0$. (B) $\lambda_2 := \lambda_2(\sigma) \ne 0$.*

Part (A) of this condition was introduced by Ghorbani et al. (2019) to simplify the analysis of the test error of the random features model $f_{\mathrm{RF}}$. Part (B) ensures that the random features model $f_{\mathrm{RF}}$ does not degenerate to the null predictor.

**Definition 4.1.** *With $z \sim N(0, 1)$, define the following scalars*

$$\begin{aligned} \overline{\lambda} := \mathbb{E}[\sigma(z)^2] - \lambda_1^2, \; \kappa := \lambda_2^2 \|\Gamma\|_F^2 d / 2, \; \tau := \lambda_2 \mathrm{tr}(B\Gamma)\sqrt{d}, \\ \overline{\lambda'} := \mathbb{E}[\sigma'(z)^2] - \lambda_1^2, \; \kappa' := \lambda_3^2 \|\Gamma\|_F^2 d / 2. \end{aligned} \tag{18}$$

These coefficients will turn out to be "sufficient statistics" which will completely capture the influence of activation function $\sigma$ on the robustness of the random features model $f_{\mathrm{RF}}$. Note that by construction, $\overline{\lambda}, \kappa, \overline{\lambda'}$, and $\kappa'$ are nonnegative. Now, consider the random psd matrices $A_0$ and $D_0$ defined by

$$A_0 := \overline{\lambda} I_m + \lambda_1^2 \Theta, \; D_0 := \overline{\lambda'} I_m + (\kappa'/d + \lambda_1^2)\Theta, \tag{19}$$

with $\Theta := WW^\top \in \mathbb{R}^{m \times m}$. These matrices appear upon linearizing the expressions for the test error and the robustness error of the RF model, using RMT techniques from El Karoui (2010). By employing the so-called *Silverstein fixed-point equation* (Silverstein & Choi, 1995; Ledoit & Péché, 2011; Dobriban & Wager, 2018), one can show that there exist positive constants $\psi_1$ and $\psi_2$ such that

$$\mathrm{tr}(A_0^{-1})/d \xrightarrow{a.s} \psi_1, \; \mathrm{tr}(A_0^{-2} D_0)/d \xrightarrow{a.s} \psi_2 \text{ in the limit (3).} \tag{20}$$

Moreover, the $\psi_k$'s only depend on (i) the parametrization rate $\rho$, and (ii) the limiting eigenvalue distribution $\mathcal{D}$ of the rescaled covariance matrix $d \cdot \Gamma$ of the hidden neurons at initialization. Also, since $\overline{\lambda}$ and $\overline{\lambda'}$ are strictly positive (thanks to Condition 4.2), so are the $\psi_k$'s by definition of $D_0$ and $A_0$ in (19). These scalars together with those defined in (18) will play a crucial role in our analysis.

### 4.2 TEST ERROR / PREDICTION PERFORMANCE IN RANDOM FEATURES REGIME

We recall that the (normalized) test error $\widetilde{\varepsilon}_{\text{test}}(f_{\text{RF}})$ of the random features model $f_{\text{RF}}$ was completely analyzed in Ghorbani et al. (2019). Indeed, the following was established that

$$\widetilde{\varepsilon}_{\text{test}}(f_{\text{RF}}) = 1 - \frac{\psi_1 \tau^2}{\|B\|_F^2 (2\kappa\psi_1 + 2)} + o_{d,\mathbb{P}}(1), \text{ in the limit (3).} \quad (21)$$

See Theorem 1 of the said paper. Thus, the normalized test error only depends on the aspect ratio $\rho$, the limiting spectral distribution $\mathcal{D}$ of $d \cdot \Gamma$, and the scale parameters $(\overline{\lambda}, \kappa, \tau)$ defined in (18). It was further that, if the task-alignment $\alpha$ of the hidden neurons defined in (17), admits a limit $\alpha_\infty$ when $d \to \infty$, then w.p 1

$$\lim_{\substack{\rho \to \infty}} \lim_{\substack{m,d \to \infty \\ m/d \to \rho}} \widetilde{\varepsilon}_{\text{test}}(f_{\text{RF}}) = 1 - \alpha_\infty^2. \quad (22)$$

Thus, (21) predicts that the normalized test error $\widetilde{\varepsilon}_{\text{test}}(f_{\text{RF}})$ vanishes if (i) $\Gamma \propto B$ and (ii) the number of neurons per input dimension $m/d$ diverges, corresponding to extreme over-parametrization.

### 4.3 ANALYSIS OF ROBUSTNESS IN RANDOM FEATURES REGIME

The following result establishes an analytic formula for the robustness in the RF regime.

**Theorem 4.1.** *Consider the random features model $f_{\text{RF}}$ (13), with covariance matrix $\Gamma$ satisfying Condition 4.1 and activation function $\sigma$ satisfying Conditions 4.2 and 4.3.*

*(A) In the limit (3), we have the following approximation*

$$\widetilde{\varepsilon}_{\text{rob}}(f_{\text{RF}}) = \frac{\tau^2 (2\kappa\psi_1^2 + \psi_2)}{\|B\|_F^2 (2\kappa\psi_1 + 2)^2} + o_{d,\mathbb{P}}(1). \quad (23)$$

*(B) Moreover, if $\lim_{d \to \infty} \alpha = \alpha_\infty$, then $\lim_{\substack{\rho \to \infty}} \lim_{\substack{m,d \to \infty \\ m/d \to \rho}} \widetilde{\varepsilon}_{\text{rob}}(f_{\text{RF}}) = \alpha_\infty^2$ w.p 1. In particular, for the*

*optimal choice of $\Gamma$ in terms of test error, namely $\Gamma \propto B$, one has $\lim_{\substack{\rho \to \infty}} \lim_{\substack{m,d \to \infty \\ m/d \to \rho}} \widetilde{\varepsilon}_{\text{rob}}(f_{\text{RF}}) = 1$ w.p 1.*

Thus, the robustness only depends on the aspect ratio $\rho$, the limiting spectral distribution $\mathcal{D}$ of $d \cdot \Gamma$ (via $\psi_1$ and $\psi_2$), and the scale parameters defined in (18). The theorem is proved in the Appendix G.4.

**Tradeoff between approximation and robustness.** We deduce from the above theorem that in the limit (3), the random features model $f_{\text{RF}}$ is more robust (i.e., less sensitive to perturbations) than the teacher model $f_\star$. Interestingly, we see that this gap in robustness between the two models closes with increasing alignment $\alpha$ between the covariance matrix of the random features $\Gamma$ and the coefficient matrix $B$. Comparing with (22), we obtain the following relationship (provided $\|\Gamma\|_F^2 \gg 1/m$),

$$\widetilde{\varepsilon}_{\text{test}}(f_{\text{RF}}) + \widetilde{\varepsilon}_{\text{rob}}(f_{\text{RF}}) \xrightarrow{p} 1 - \alpha_\infty^2 + \alpha_\infty^2 = 1, \text{ in the limit (3),} \quad (24)$$

which trades-off between the normalized test error $\widetilde{\varepsilon}_{\text{test}}(f_{\text{RF}})$ (defined in (21)) and the normalized robustness $\widetilde{\varepsilon}_{\text{rob}}(f_{\text{RF}})$ of $f_{\text{RF}}$. Thus, we have identified another novel tradeoff between the test error and the robustness in random features models.

Theorem 4.1 and Corollary H.1 are empirically verified in Fig. 1. Results for the ReLU activation function are also shown. Notice the perfect match between our theoretical results and experiments.

## 5 NEURAL TANGENT REGIME

Consider a two-layer network width output weights $z_j^0$ fixed to 1, and hidden weight $w_j \in \mathbb{R}^d$ drawn from $N(0, \Gamma)$. For the quadratic activation $\sigma(t) := t^2$, the neural tangent (NT) approximation (Jacot et al., 2018; Chizat et al., 2019) w.r.t. the first layer parameters is given by

$$f_{W+A}(x) \approx f_{\text{init}}(x) + \text{tr}(A \nabla_W f_W(x)) = f_{\text{init}}(x) + 2 \sum_{j=1}^m (x^\top (z_j^0 a_j))(x^\top w_j). \quad (25)$$

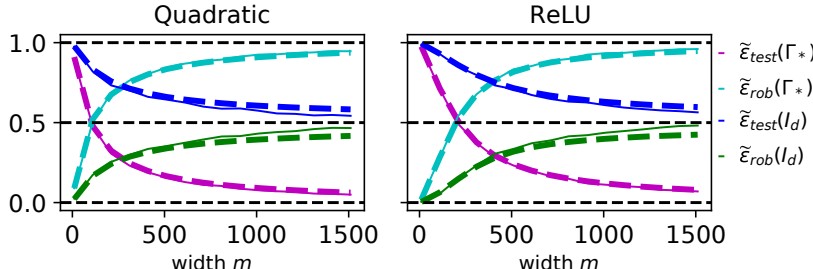

Figure 1: Empirical validation of Theorem 4.1 and Corollary H.1. Showing (normalized) test / test error $\widetilde{\varepsilon}_{\text{test}}$ and robustness $\widetilde{\varepsilon}_{\text{rob}}$ of random features model (13) as a function of the number of the network width, for different choices of the covariance matrix $\Gamma$ of the random weights of the hidden neurons: the optimal choice $\Gamma_\star \propto B$ and the naive choice $I_d$. Here, the input-dimension is $d = 450$ and the regularization $\lambda$ is zero. Horizontal broken lines correspond to asymptotes at $\alpha_\infty^2$ at $1 - \alpha_\infty^2$, where $\alpha_\infty := \lim_{d \to \infty} \text{tr}(B\Gamma)/(\|B\|_F \|\Gamma\|_F)$ is the level of task-alignment of the covariance matrix $\Gamma$ of hidden neurons, w.r.t learning the teacher model $f_\star$ defined in (1). Broken curves are theoretical predictions, while solid curves correspond to actual experiments.

where $f_{\text{init}}$ is the function computed by the neural network at initialization (see Appendix E for details), and $A = (a_1, \ldots, a_m) = (\Delta w_1, \ldots, \Delta w_m) \in \mathbb{R}^{m \times d}$ is the change in $W$. We will see that the initialization term $f_{\text{init}}$ might have drastic influence on the robustness of the resulting model.

## 5.1 NEURAL TANGENT APPROXIMATION WITHOUT INITIALIZATION TERM

We temporarily discard the initialization term $f_{\text{init}}(x)$ from the RHS of (25), and consider the simplified approximation

$$f_{\text{NT}}(x; A; c) := 2 \sum_{j=1}^{m} (x^\top a_j)(x^\top w_j) - c, \tag{26}$$

where, WLOG, we absorb the output weights $z_j$ in the parameters $a_j$ in the first-order term. In (26), $A \in \mathbb{R}^{m \times d}$ and $c \in \mathbb{R}$ are model parameters that are optimized. In terms of test error, let $A_{\text{NT}}$ and $c_{\text{NT}}$ be optimal in $f_{\text{NT}}(\cdot; A, c)$, and let $f_{\text{NT}} = f_{\text{NT}}(\cdot; A_{\text{NT}}, c_{\text{NT}})$ for short. In Thm. 2 of Ghorbani et al. (2019), it is shown that the (normalized) test error of the linearized model $f_{\text{NT}}$ is given by

$$\mathbb{E}_W[\widetilde{\varepsilon}_{\text{test}}(f_{\text{NT}})] = (1 - \rho)_+^2 (1 - \beta) + (1 - \rho)_+ \beta + o_d(1). \tag{27}$$

where $\beta = \beta(B) := \text{tr}(B)^2/(d\|B\|_F^2) \in (0, 1]$. We now establish an analytic formula for the robustness error of $f_{\text{NT}}$.

**Theorem 5.1.** *Consider the neural tangent model $f_{\text{NT}}$ in (26). In the limit (3) it holds that,*

$$\mathbb{E}_W[\widetilde{\varepsilon}_{\text{rob}}(f_{\text{NT}})] = (\underline{\rho} + \underline{\rho}^2)/2 + (\underline{\rho} - \underline{\rho}^2)\beta/2 + o_d(1), \text{ where } \underline{\rho} := \min(\rho, 1). \tag{28}$$

Further observe that because $0 \le \beta \le 1$, the RHS of (28) is further upper-bounded by $\underline{\rho} \le 1$ with equality when $\beta = 1$ (e.g., for $B \propto I_d$). We deduce that in the NT regime, the student neural network is at least as robust as the teacher model $f_\star$. Comparing with (27), we obtain the following tradeoff between test error and robustness, stated only for $\beta = 1$ for simplicity of presentation.

**Corollary 5.1.** *If $\beta = 1$ (i.e., if $B \propto I_d$), then in the limit (3), it holds that*

$$\mathbb{E}_W[\widetilde{\varepsilon}_{\text{test}}(f_{\text{NT}}) + \widetilde{\varepsilon}_{\text{rob}}(f_{\text{NT}})] = 1 + o_d(1). \tag{29}$$

## 5.2 NEURAL TANGENT APPROXIMATION WITH INITIALIZATION TERM

We now consider the neural tangent approximation (25) without discarding the initialization term $f_{\text{init}}$ from the RHS of (25). Also, let $z^0 \in \mathbb{R}^m$ be the output weights, drawn iid from $N(0, 1/m)$ and frozen, and let $Q$ be the $m \times m$ diagonal matrix with $z^0$ as its diagonal. This corresponds to what

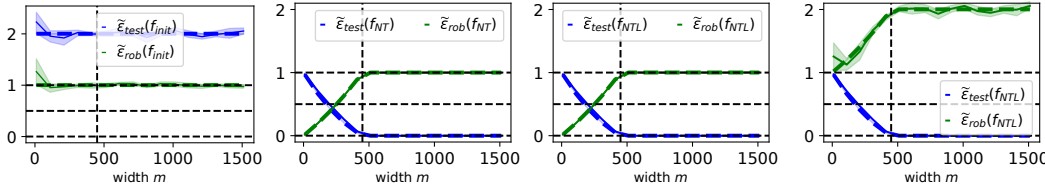

(a) NN at large random init     (b) NT regime     (c) Lazy, small random init     (d) Lazy, large random init

Figure 2: Showing curves of (normalized) test error $\widetilde{\varepsilon}_{\text{test}}$ and robustness $\widetilde{\varepsilon}_{\text{rob}}$ for a two-layer neural network in different learning regimes of the hidden weights. Here, the input dimension is $d = 450$ and the width $m$ sweeps a range of values from $10$ to $1500$. Dashed curves correspond to theoretical predictions, while solid curves correspond to actual values observed in the experiment (5 runs). We use $n = 10^6$ training samples as a proxy for infinite data. The covariance matrix of the hidden neurons is fixed at $\Gamma = (1/d)I_d$. **For simplicity of this experiment**, we also take the coefficient matrix $B$ of the teacher model to be proportional to $I_d$. **(c)** "Small random init" means $B = (1/\sqrt{d})I_d$, so that $B$ is much larger than $\Gamma$. **(d)** "Large random init" means that $B = (1/d)I_d$, and thus is of the same order as $\Gamma$ (in Frobenius norm). In this case, the initialization degrades the robustness, as predicted by Thm. 5.2. Note that, as predicted by Thm. 5.2, random initialization has no impact on the test error of the NT approximation. Results for **(a)** the neural network at initialization (Thm. E.1 and E.2) and **(b)** In the NT regime (Thm. 5.1) are also depicted for reference.

could be referred to as *lazy training* Chizat et al. (2019) regime of the hidden layer. Let $f_{\text{NTL}}(x; A, c)$ be RHS of (25),

$$f_{\text{NTL}}(x; A, c) := f_{W,z,c}(x) + 2 \sum_{j=1}^{m} (x^\top w_j)(x^\top a_j) = f_{\text{init}}(x) + f_{\text{NT}}(x; A, c), \quad (30)$$

where $f_{\text{init}}(x) := \sum_{j=1}^{m} z_j^0 (x^\top w_j)^2 = x^\top W^\top Q W x$ defines the neural network at initialization.

**Theorem 5.2.** *Suppose the output weights $z^0$ at initialization are iid from $N(0, (1/m)I_m)$. Then, in the limit (3), the following identities hold*

$$\mathbb{E}_{\{W,z^0\}}[\widetilde{\varepsilon}_{\text{test}}(f_{\text{NTL}})] = \mathbb{E}_W[\widetilde{\varepsilon}_{\text{test}}(f_{\text{NT}})] + o_d(1), \quad (31)$$

$$\mathbb{E}_{\{W,z^0\}}[\widetilde{\varepsilon}_{\text{rob}}(f_{\text{NTL}})] = \mathbb{E}_W[\widetilde{\varepsilon}_{\text{rob}}(f_{\text{NT}})] + \mathbb{E}_{\{W,z^0\}}[\widetilde{\varepsilon}_{\text{rob}}(f_{\text{init}})] + o_d(1). \quad (32)$$

Thus, on average (over initialization): (i) $f_{\text{NTL}}$ and $f_{\text{NT}}$ have the same test error, i.e., the initialization term $f_{\text{init}}$ in $f_{\text{NTL}}$ does not affect its test error. (ii) On the other hand, $f_{\text{NTL}}$ is less robust than $f_{\text{NT}}$; the deficit in robustness, namely the term $(1 + \|\Gamma\|_F^2)/\|B\|_F^2$, corresponds exactly to the contribution of the initialization. The situation is empirically illustrated in Fig. 2. Notice the perfect match between our theoretical results and experiments.

## 6 CONCLUDING REMARKS

In this paper, we have studied the adversarial robustness of two-layer neural networks in different high-dimensional learning regimes, and established a number of new tradeoffs between prediction performance and robustness, in the form (8). Our analysis also shows that random initialization can further degrade the robustness in lazy training regimes: for "large" random initialization, the trained neural network inherits additional vulnerability already present at initialization. Our work can be seen as a first step towards a rigorous theoretical understanding of the robustness of trained neural networks, an important subject which is still understudied.

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

# Appendix / Supplementary Materials

CONTENTS

# A   FURTHER RELATED WORK

Various works have theoretically studied adversarial examples and robustness in supervised learning, and the relationship to ordinary predictive performance / test error. We present a detailed list here.

Tsipras et al. (2019) considers a specific data distribution where good accuracy implies poor robustness. Shafahi et al. (2018); Mahloujifar et al. (2018); Gilmer et al. (2018); Dohmatob (2019) show that for high-dimensional data distributions which have concentration property (e.g., multivariate Gaussians, distributions satisfying log-Sobolev inequalities, etc.), an imperfect classifier will admit adversarial examples. Dobriban et al. (2020) studies tradeoffs in Gaussian mixture classification problems, highlighting the impact of class imbalance. On the other hand, Yang et al. (2020) observed empirically that natural images are well-separated, and so locally-lipschitz classifies shouldn't suffer any kind of test error vs robustness tradeoff. However, gradient-descent is not likely to find such models. Our work studies regression problems with quadratic targets, and shows that there are indeed tradeoffs between test error and robustness which are controlled by the learning algorithm / regime and model.

Simon-Gabriel et al. (2019); Daniely & Shacham (2020); Bubeck et al. (2021); Bartlett et al. (2021) study adversarial vulnerability of neural networks at initialization, but do not consider the effects of training the model, in contrast to our work.

Schmidt et al. (2018); Khim & Loh (2018); Yin et al. (2019); Bhattacharjee et al. (2021); Min et al. (2021b;a) study the sample complexity of robust learning. In contrasts, our work focuses on the case of infinite data, so that the only complexity parameters are the input dimension $d$ and the network width $m$. Bhattacharjee et al. (2021) studies robustness vs accuracy for data distributions which are well-separated (e.g., say the two classes are supported on disjoint balls). The main finding in that paper is that (i) the robustness vs accuracy tradeoff doesn't exist for well-separated datasets. The work also posits that (ii) real-world datasets are well-separated. We think (i) is only an artifact of the well-separatedness assumption (an assumption which fails for Gaussians (as noted in the paper), say, due to infinite support). Also, (ii) is likely due to the fact that most real datasets are limited in sample size, and so, deceptively appear to be well-separated. Indeed, in the real world, there are cats which look like dogs (e.g, Siamese cats), even though such data might be under-represented in ML datasets.

Gao et al. (2019); Bubeck et al. (2020b); Bubeck & Sellke (2021) show that over-parameterization may be necessary for robust interpolation in the presence of noise. In contrast, our paper considers a structured problem with noiseless signal and infinite-data $n = \infty$, where the network width $m$ and the input dimension $d$ tend to infinity proportionately. In this under-complete asymptotic setting, our results show a precise picture of the tradeoffs between approximation (test error) and robustness in different learning regimes. Our work nuances this picture by exhibiting a nontrivial interplay between robustness and test error which persists even in the case of infinite samples, where the model isn't affected by label noise.

# B   NOTATIONS

**Linear algebra.**   The set of integers from $1$ through $d$ will be denoted $[d]$. We will denote the identity matrix of size $d$ by $I_d$. The euclidean norm of a vector $x \in \mathbb{R}^d$ will be denoted $\|x\|$. The $k$th largest singular-value of a matrix $A$ will be denoted $s_k(A)$, and equals the positive square-root of the $k$th largest eigenvalue of the positive-semidefinite (psd) matrix $AA^\top$. In particular, $\|A\|_{op} := s_1(A)$ is the spectral norm of $A$. If $A$ is itself psd, then its singular-values coincide with its eigenvalues. The Frobenius norm of $A$ is denoted $\|A\|_F$ and defined by $\|A\|_F := \sqrt{\sum_{k=1}^d s_k(A)^2}$. More generally, we define $\|A\|_{F,m} := \sqrt{\sum_{k=1}^{m \wedge d} s_k(A)^2}$, so that $\|A\|_{F,d} = \|A\|_F$ in particular. Note that $m \mapsto \|A\|_{F,m}$ is a nondecreasing function which is upper-bounded by $\|A\|_F$. The Hadamard / element-wise product of two matrices $A_1$ and $A_2$ of the same shape will be denoted $A_1 \circ A_2$. The squared $L_2$-norm of a function $f : \mathbb{R}^d \to \mathbb{R}$ w.r.t a distribution $P$ on $\mathbb{R}^d$ will be denoted $\|f\|_P^2$, and defined by $\|f\|_{L^2(P)} := \mathbb{E}_P[f(x)^2]$, whenever this integral exists. Given a psd matrix of size $d$, we denote by $N(0, \Sigma)$ the $d$-dimensional multivariate gaussian distribution with covariance matrix $\Sigma$.

**Asymptotics.**   The usual notation $\mathcal{O}_d(1)$ (resp. $\mathcal{O}_{d,\mathbb{P}}(1)$) is used to denote a quantity which remains bounded (resp. bounded in probability) in the limit $d \to \infty$. Likewise $o_d(1)$ (resp. $o_{d,\mathbb{P}}(1)$) denotes a

quantity which goes to zero (resp. which goes to zero in probability) in the limit $d \to \infty$. As usual, the acronym "a.s" stands for *almost-surely*, while "w.p $p$" stands for *with probability at least $p$.*

## C  WARM-UP: AN INSIGHT FROM LINEAR REGRESSION

Before providing complete proofs in the sequel, in this section we will use linear regression to develop an intuitive understanding for the tradeoffs established in the paper.

Consider $n$ sample points $(x_1, y_1), \ldots, (x_n, y_n)$ in $\mathbb{R}^d \times \mathbb{R}$. Now, let $X = (x_1, \ldots, x_n) \in \mathbb{R}^{n \times d}$ be the design matrix and $y := (y_1, \ldots, y_n) \in \mathbb{R}^n$ be the response vector. For any nonempty subset $S$ of $[n]$, let $X_S \in \mathbb{R}^{|S| \times d}$ (resp. $y_S \in \mathbb{R}^{|S|}$) be the version of $X$ (resp. $y$) with all rows (resp. columns) not in $S$ removed. For a (possibly random) sequence of nonempty subsets $(S_t)_{t=1}^\infty$ of $[n]$, and a sequence of sufficiently small stepsizes $(\alpha_t)_{t=1}^\infty$, the following discrete dynamics

$$w_t = w_{t-1} - \alpha_t X_{S_t}^\top (X_{S_t} w_{t-1} - y_{S_t}) / |S_t| \tag{33}$$

represents GD or SGD initialized at $w_0 \in \mathbb{R}^d$. In particular, GD corresponds to taking $S_t = [n]$ for all $t \geq 1$. By construction, it is clear from (33) that at any iteration $t$,

$$w_t - w_0 \in \text{span}(X) := \{X^\top z \mid z \in \mathbb{R}^n\}. \tag{34}$$

On the other hand, suppose $w_\star \in \mathbb{R}^d$ is an interpolant, i.e $Xw_\star = y$. It is well-known that $w_t$ converges to a point $w_\infty$ which is the orthogonal projection of $w_0$ onto the affine space of interpolants $\mathcal{I} := \{w \in \mathbb{R}^d \mid Xw = y\} = w_\star + \text{kern}(X)$, where $\text{kern}(X) \subseteq \mathbb{R}^d$ is the kernel of $X$. Thanks to (34) and the closedness of the subspace $\text{span}(X)$, it is clear that $w_\infty - w_0 \in \text{span}(X)$. We conclude that $w_\infty - w_\star$ is orthogonal to $w_\infty - w_0$, and so $\|w_\star - w_0\|^2 = \|w_\star - w_\infty\|^2 + \|w_\infty - w_0\|^2$, by *Pythagoras' Theorem*. In particular, taking $w_0 = 0$, the previous identity becomes

$$\underbrace{\|w_\infty - w_\star\|^2}_{\text{test error of } w_\infty} + \underbrace{\|w_\infty\|^2}_{\text{rob. error of } w_\infty} = \underbrace{\|w_\star\|^2}_{\text{constant}}, \tag{35}$$

where the test error is w.r.t to test data generated according to the linear model $x \sim P_x := N(0, I_d)$, $y = f_{w_\star}(x)$, with $f_w(x) := x^\top w$ for all $w, x \in \mathbb{R}^d$. Dividing through by $\|w_\star\|^2 = \|f_{w_\star}\|_{L^2(P_x)}^2$, we deduce the following result which can be seen as the inductive bias of GD and SGD on linear regression.

**Proposition C.1.** *For GD or SGD started at $w_0 = 0$, it holds that $\widetilde{\varepsilon}_{\text{test}}(f_{w_\infty}) + \widetilde{\varepsilon}_{\text{rob}}(f_{w_\infty}) = 1$.*

This is a tradeoff between the test error and the robustness error for GD and SGD! It is valid for all sample sizes $n \geq 1$. In contrast, the tradeoffs established in the nonlinear settings of the previous sections persist hold for infinite samples where $n = \infty$. Perhaps, sufficiently large but finite $n$ is sufficient, but this investigation is left for future work.

## D  JUSTIFICATION OF OUR PROPOSED MEASURE OF ROBUSTNESS

Let us begin by explaining why our proposed measure of robustness based on Dirichlet energy (6) is actually a measure of robustness.

Unless otherwise stated, in this section the feature distribution will be any distribution $P_x$ on $\mathbb{R}^d$. Given smooth[1] $f : \mathbb{R}^d \to \mathbb{R}$, consider the $d \times d$ psd matrix $J(f)$ and scalar $\mathfrak{S}(f) \geq 0$ defined by

$$J(f) := \mathbb{E}_{P_x}[\nabla f(x) \nabla f(x)^\top], \quad \mathfrak{S}(f) := \text{tr}(J(f))^{1/2}. \tag{36}$$

Note that $\varepsilon_{\text{rob}}(f) = \mathfrak{S}(f)^2$. The following lemma shows that $\mathfrak{S}(f)$ measures the sensitivity of $f$ to random local fluctuations in test data, on average.

---

[1]Here, derivatives are allowed to be defined only almost-everywhere, as in neural networks with ReLU activation function. This notion of smoothness is completely subsumed by the more general notion presented in Section 4.1 of Gigli & Ledoux (2013).

**Lemma D.1** (Measure of local sensitivity). *We have*

$$\lim_{\delta \to 0^+} \frac{1}{\delta} \mathbb{E}_{P_x}[\Delta_f(x; \delta)] = \mathfrak{S}(f), \tag{37}$$

*where $\Delta_f(x; \delta) := \sup_{\|v\|_2 \le \delta} |f(x + v) - f(x)|$.*

This lemma is a direct corollary to Lemma D.3 proved later below.

The next lemma shows that $\|J(f)\|_{op}$ measures the (non)robustness of $f$ to universal adversarial perturbations, in the sense of Moosavi-Dezfooli et al. (2017).

**Lemma D.2** (Measure of robustness to universal perturbations). *We have the identity*

$$\lim_{\delta \to 0^+} \frac{1}{\delta} \Delta_f(\delta) = \|J(f)\|_{op}^{1/2}, \tag{38}$$

*where $\Delta_f(\delta)^2 := \sup_{\|v\| \le \delta} \mathbb{E}_{P_x}(f(x + v) - f(x))^2$.*

In particular, the leading eigenvector of $J(f)$ corresponds to (first-order) universal adversarial perturbations of $f$, in the sense of Moosavi-Dezfooli et al. (2017), which can be efficiently computed using the *Power Method*, for example.

A rough sketch of the proof of the above lemma is as follows. To first-order, we have $f(x+v) - f(x) \approx v^\top \nabla f(x)$. Thus,

$$
\begin{aligned}
\Delta_f(\delta)^2 &:= \sup_{\|v\| \le \delta} \mathbb{E}_{P_x}[(f(x + v) - f(x))^2] \\
&\approx \sup_{\|v\| \le \delta} \mathbb{E}_{P_x}[(v^\top \nabla f(x))^2] \\
&= \sup_{\|v\| \le \delta} v^\top J(f) v = \delta^2 \|J(f)\|_{op}^2.
\end{aligned}
$$

The first lemma is proved via a similar argument.

### D.1 WHY NOT USE LIPSCHITZ CONSTANTS TO MEASURE ROBUSTNESS ?

Note for any that smooth function, $\mathfrak{S}(f)$ is always a lower-bound for the Lipschitz constant $\|f\|_{\mathrm{Lip}}$ of $f$. Recall that $\|f\|_{\mathrm{Lip}}$ is defined by

$$\|f\|_{\mathrm{Lip}} := \sup_{x \ne x'} \frac{|f(x) - f(x')|}{\|x - x'\|}. \tag{39}$$

One special case where there is equality $\mathfrak{S}(f) = \|f\|_{\mathrm{Lip}}$ is when $f$ is a linear function. However, this is far from true in general: $\|f\|_{\mathrm{Lip}}$ is a worst-case measure, while $\mathfrak{S}(f)$ is an average-case measure for each $q$. If $\|f\|_{\mathrm{Lip}}$ is small (i.e., of order $\mathcal{O}(1)$), then a small perturbation (i.e., of size $\mathcal{O}(1)$) can only result in mild change in the output of $f$ (i.e., of order $\mathcal{O}(1)$). However, a large value of $\|f\|_{\mathrm{Lip}}$ is uninformative regarding adversarial examples (for example, one can think of a function which is smooth everywhere except on a set of measure zero). In contrast, a large value for $\mathfrak{S}(f)$ indicates that, on average, it is possible for an adversarial to drastically change the output of $f$ via a small modification of its input.

**An illustrative example.** Consider a quadratic function $f(x) := (1/2)x^\top B x + c$ with isotropic feature distribution $P_x = N(0, I_d)$. Note that the teacher model $f_\star$ defined in (1) is of this form. A direct computation reveals that $\nabla f(x) = Bx$ and so $\mathfrak{S}(f)^2 := \mathbb{E}_{P_x} \|\nabla f(x)\|^2 = \mathbb{E}_{P_x} \|Bx\|^2 = \|B\|_F^2$. However, the Lipschitz constant of $f$ restricted to the ball of radius $\sqrt{d}$ is[2],

$$\|f\|_{\widetilde{\mathrm{Lip}}} = \sup_{\|x\| \le \sqrt{d}} \|\nabla f(x)\| = \sup_{\|x\| \le \sqrt{d}} \|Bx\| = \sqrt{d} \|B\|_{op},$$

which can be up to $\sqrt{d}$ times larger than $\mathfrak{S}(f) = \|B\|_F$. For example, take $B$ to be an ill-conditioned, e.g., rank-1, matrix.

---

[2]For fair comparison with our measure of robustness, we restrict the computation of Lipschitz constant to this ball since $\sqrt{d}$ is the length of a typical random vector from $N(0, I_d)$.

### D.2 PROOFS FOR DIRICHLET ENERGY AS A MEASURE OF ADVERSARIAL VULNERABILITY

Let $\|\cdot\|$ be any norm on $\mathbb{R}^d$ with dual norm $\|\cdot\|_\star$. Given a function $f : \mathbb{R}^d \to \mathbb{R}$, a tolerance parameter $\delta \geq 0$ (the *attack budget*), and a scalar $q \geq 1$, define $R_\delta(f)$ by

$$R_{q,\delta}(f,g) := \mathbb{E}_{P_x}\left[\Delta_f(x;\delta)^q\right], \tag{40}$$

where $\Delta_f(x;\delta) := \sup_{\|x'-x\|\leq\delta}|f(x') - f(x)|$ is the maximal variation of $f$ in a neighborhood of size $\delta$ around $x$. For $q = 2$, we simply write $R_\delta(f,g)$ for $R_{\delta,2}(f,g)$. In particular, $G_\delta(f) := \mathbb{E}[R_\delta(f, f_\star)]$ is *adversarial test error* and $G_0(f) := \mathbb{E}[R_0(f, f_\star)]$ is the ordinary *test error* of $f$, where the expectations are w.r.t all sources of randomness in $f$ and $f_\star$. Of course $G_\delta(f)$ is an increasing function of $\delta$.

Define $R_{q,\delta}(f) := R_{q,\delta}(f, f)$ and $R_\delta(f) := R_{2,\delta}(f, f)$, which measure the deviation of the outputs of $f$ w.r.t to the outputs of $f$, under adversarial attack. Note $R_q(f) \equiv 0$. Also note that in the case where $\|\cdot\|$ is the euclidean $L_2$-norm: if $f$ is a near perfect model (in the classical sense), meaning that its ordinary test error $G_0(f)$ is small, then $R_\delta(f)$ is a good approximation for $G_\delta(f)$. Finally, (at least for small values of $\alpha$), we can further approximate $R_\alpha(f)$ (and therefore $G_\delta(f)$, for near perfect $f$) by $\delta^2$ times the Dirichlet energy $\mathfrak{S}(f)^2$. Indeed,

**Lemma D.3.** *Let* $q \in [1, \infty)$*, and* $f : \mathbb{R}^d \to \mathbb{R}$ *be a smooth function. Define* $\mathfrak{S}_q(f)$ *by*

$$\mathfrak{S}_q(f) := \left(\mathbb{E}_{P_x}[\|\nabla f(x)\|_\star^q]\right)^{1/q}. \tag{41}$$

*Note that, in particular, if* $\|\cdot\|$ *is the euclidean* $L_2$*-norm and* $q = 2$*, then* $\mathfrak{S}_q(f)^2$ *is the Dirichlet energy defined in* (6) *as our measure of robustness. We have the following*

*(A) **General case.*** $\mathfrak{S}_q(f)$ *is the right derivative of the mapping* $\delta \mapsto R_\delta(f)^{1/q}$ *at* $\delta = 0$*. More precisely, we have the following*

$$\lim_{\delta \to 0^+} \frac{R_{q,\delta}(f)^{1/q}}{\delta} = \mathfrak{S}_q(f), \tag{42}$$

*or equivalently,* $R_{q,\delta}(f) = \delta^q \cdot \mathfrak{S}(f)^q +$ *Higher order terms in* $\delta^q$*.*

*(B) **Case of Dirichlet energy** In particular, if* $\|\cdot\|$ *is the euclidean* $L_2$*-norm, and we take* $q = 2$*,*

$$R_\delta(f) = \delta^2 \cdot \mathfrak{S}_q(f)^2 + \text{ Higher order terms in } \delta^2. \tag{43}$$

**Remark D.1.** *A heuristic argument was used in* Simon-Gabriel et al. *(2019) to justify the use of average (dual-)norm of gradient (i.e the average local Lipschitz constant)* $\mathbb{E}_{P_x}[\|\nabla f(x)\|_\star]$ *(corresponding to* $q = 1$ *in the above) as a proxy for the adversarial generalization.*

The proof of Lemma D.3 follows directly *Fubini's Theorem* and the following lemma.

**Lemma D.4.** *If* $f$ *is differentiable at* $x$*, then the function* $\delta \mapsto \Delta_f(x;\delta) := \sup_{\|x'-x\|\leq\delta}|f(x') - f(x)|$ *is right-differentiable at* $0$ *with derivative given by* $\Delta'_f(x;0) = \|\nabla f(x)\|_\star$*.*

*Proof.* As $f$ is differentiable, $f(x') = f(x) + \nabla f(x)^\top(x' - x) + o(\|x' - x\|)$ around $x$. Therefore for sufficiently small $\delta$, if $B(x;\delta)$ is ball of radius $\delta$ around $x$, then

$$\begin{aligned}
\Delta_f(x;\delta) &= \sup_{x' \in B(x;\delta)} |\nabla f(x)^\top(x' - x) + o(\|x' - x\|)| \\
&\leq \sup_{x' \in B(x;\delta)} |\nabla f(x)^\top(x' - x)| + \sup_{y \in B(x;\delta)} o(\|x' - x\|) \\
&= \|\nabla f(x)\|_\star \delta + \sup_{y \in B(x;\delta)} o(\|x' - x\|) \frac{\Delta_f(x;\delta)}{\delta} \\
&\leq \|\nabla f(x)\|_\star + \sup_{x' \in B(x;\delta)} \frac{o(\|x' - x\|)}{\delta}
\end{aligned} \tag{44}$$

Note that $\sup_{x' \in B(x;\delta)} \frac{o(\|x' - x\|)}{\delta} \to 0$. This proves $\limsup_{\delta \to 0^+} (1/\delta)\Delta_f(x;\delta) \leq \|\nabla f(x)\|_\star$. Similarly, one computes

$$
\begin{aligned}
\Delta_f(x;\delta) &= \sup | \nabla f(x)^\top (x' - x) + o(\|x' - x\|) | \\
&\geq \sup |\nabla f(x)^\top (x' - x)| - \sup o(\|x' - x\|) \\
&= \|\nabla f(x)\|\delta - \sup o(\|x' - x\|)
\end{aligned}
\tag{45}
$$

Hence $\liminf_{\delta \to 0^+} (1/\delta)\Delta_f(x;\delta) \geq \|\nabla f(x)\|_\star$, and we conclude that $\delta \mapsto \Delta_f(x;\delta)$ is differentiable at $\delta = 0$, with derivative $\Delta'_f(x;0) = \|\nabla f(x)\|_\star$ as claimed. □

*Proof of Lemma D.3.* By basic properties of limits, one has

$$
\begin{aligned}
\left( \lim_{\delta \to 0^+} \frac{R_{q,\delta}(f)^{1/q}}{\delta^q} \right)^q &= \lim_{\delta \to 0^+} \frac{R_{q,\delta}(f)}{\delta} \\
&= \lim_{\delta \to 0^+} \frac{\mathbb{E}_{P_x}[|\Delta_f(x;\delta)|^q]}{\delta} \\
&= \mathbb{E}_{P_x}\left[ \lim_{\delta \to 0^+} \frac{|\Delta_f(x;\delta)|^q}{\delta} \right] \\
&= \mathbb{E}_{P_x}\left[ \left( \lim_{\delta \to 0^+} \frac{|\Delta_f(x;\delta)|}{\delta} \right)^q \right] \\
&= \mathbb{E}_{P_x}[\|\nabla f(x)\|_\star^q] \\
&:= \mathfrak{S}_q(f)^q,
\end{aligned}
\tag{46}
$$

where the 3rd line is thanks to *Fubini's Theorem*, and the 5th line is thanks to lemma D.4 (and the fact that $\Delta_f(x;0) \equiv 0$). Noting that $R_{q,0}(f) \equiv 0$ then concludes the proof. □

# E  NEURAL NETWORKS AT (RANDOM) INITIALIZATION

We now consider networks at initialization, wherein the hidden weights matrix $W = (w_1, \ldots, w_m)$ is a random $m \times d$ matrix with iid rows from $N(0, \Gamma)$ as in the random features regime (13), but we freeze the output weight vector $z = z_0 \in \mathbb{R}^m$ at random initialization, with random iid entries from $N(0, 1/m)$, following standard initialization procedures. Let $f_{\text{init}}$ denote this random network, i.e.,

$$
f_{\text{init}}(x) := (z^0)^\top \sigma(Wx) = \sum_{j=1}^m z_j^0 \sigma(x^\top w_j).
\tag{47}
$$

**Theorem E.1.** *Under the Conditions 4.1 and 4.2, we have the identity in the limit* (3)*,*

$$
\widetilde{\varepsilon}_{\text{rob}}(f_{\text{init}}) = \frac{\|\sigma'\|_{L^2(N(0,1))}^2 + \lambda_3^2\|\Gamma\|_F^2/2 + \lambda_2^2\|\Gamma\|_F^2}{4\|B\|_F^2} + o_{d,\mathbb{P}}(1),
$$

*where $\lambda_k$ is the kth Hermite coefficient of the activation function $\sigma$. In particular, for the quadratic activation function $\sigma(t) = t^2 - 1$, we have $\widetilde{\varepsilon}_{\text{rob}}(f_{\text{init}}) = \frac{1 + \|\Gamma\|_F^2}{\|B\|_F^2} + o_{d,\mathbb{P}}(1)$.*

Analogously, the test error for the NN at initialization is given by the following result.

**Theorem E.2.** *Under the Conditions 4.1 and 4.2, we have the following identity in the limit* (3)*,*

$$
\widetilde{\varepsilon}_{\text{test}}(f_{\text{init}}) = 1 + \frac{\|\sigma\|_{L^2(N(0,1))}^2 + \lambda_2^2\|\Gamma\|_F^2/2}{2\|B\|_F^2} + o_{d,\mathbb{P}}(1).
$$

*In particular, for the quadratic activation $\sigma(t) := t^2 - 1$, we have the following identity $\widetilde{\varepsilon}_{\text{test}}(f_{\text{init}}) = 1 + \frac{1 + \|\Gamma\|_F^2}{\|B\|_F^2} + o_{d,\mathbb{P}}(1)$.*

Combining Thm. E.2 with formula (11), we deduce that training a randomly initialized neural network always improves its test error, as one would expect. On the other hand, combining Thm. 3.1 and Thm. E.1, we deduce that fully training the networks (10) via SGD:

(1) Degrades robustness if $\|B\|_F^2 \gtrsim \|\Gamma\|_F^2 + 1$. This is because in this case, the parameters of the model align to the signal matrix $B$, which has much larger energy than the parameters at initialization. Indeed, SGD tends to move the covariance structure of the hidden neurons from $\Gamma$ to $B$.

(2) Improves robustness if $\|B\|_F^2 \lesssim \|\Gamma\|_F^2 + 1$.

## F  MISCELLANEOUS

### F.1  LAZY TRAINING OF OUTPUT LAYER IN RF REGIME

We now study the influence of the initialization on the random features regime. Let $W = (w_1, \ldots, w_m) \in \mathbb{R}^{m \times d}$ with random rows drawn iid from $N(0, \Gamma)$ as in the RF model (13), and let the output layer be initialized at $z = z^0 \sim N(0, (1/m)I_d)$ and updated via single-pass gradient-flow on the entire data distribution (infinite data). In this so-called random features lazy (RFL) regime, we posit the following approximation neural network (2)

$$f_{\mathrm{RFL}}(x) := z_{\mathrm{RFL},\lambda}^\top \sigma(Wx) = f_{\mathrm{init}}(x) + \delta_\lambda^\top \sigma(Wx), \tag{48}$$

where $z_{\mathrm{RFL},\lambda} := z^0 + \delta_\lambda$ and $\delta_\lambda \in \mathbb{R}^m$ solves the following ridge-regression problem

$$\arg\min_{\delta \in \mathbb{R}^m} \mathbb{E}_{x \sim N(0,I_d)}[(\delta^\top \sigma(Wx) + f_{\mathrm{init}}(x) - f_\star(x))^2] + \lambda\|\delta\|^2. \tag{49}$$

The use of the ridge parameter here can be thought of as a proxy for early-stopping at iteration $t \propto 1/\lambda$ Ali et al. (2020); $\lambda = 0$ corresponds to training the output layer to optimality.

**Theorem F.1.** *We have the following identities*

$$\mathbb{E}_{z^0}[\widetilde{\varepsilon}_{\mathrm{test}}(f_{\mathrm{RFL},\lambda})] = \widetilde{\varepsilon}_{\mathrm{test}}(f_{\mathrm{RF}}) + \frac{\mathrm{tr}(P_\lambda^2 U)/m}{2\|B\|_F^2} + o_{d,\mathbb{P}}(1) \tag{50}$$

$$\mathbb{E}_{z^0}[\widetilde{\varepsilon}_{\mathrm{rob}}(f_{\mathrm{RFL},\lambda})] = \widetilde{\varepsilon}_{\mathrm{rob}}(f_{\mathrm{RF}}) + \frac{\mathrm{tr}(P_\lambda^2 C)/m}{4\|B\|_F^2} + o_{d,\mathbb{P}}(1), \tag{51}$$

*where $U = U(W)$ and $C = C(W)$ are the random matrices defined in* (15) *and* (54) *respectively.*

Because $P_\lambda^2$, $U$, and $C$ are psd matrices, the residual terms $\mathrm{tr}(P_\lambda^2 U)/m$ and $\mathrm{tr}(P_\lambda^2 C)/m$ in the above formulae are nonnegative. We deduce that random initialization of the output weights hurts both test error and robustness, as long as the RFL regime is valid.

**Infinitely regularized case $\lambda \to \infty$.** Note that $P_\lambda$ converges in spectral norm a.s to the identity matrix $I_m$ in the limit $\lambda \to \infty$. Thus, in this limit, $z_{\mathrm{RF},\lambda}$ converges almost-surely to the all-zero $m$-dimensional vector and so, thanks to (91), the output weights $z_{\mathrm{RFL},\lambda}$ of $f_{\mathrm{RFL},\lambda}$ converge to the value at initialization $z^0$. Therefore, $f_{\mathrm{RFL},\lambda}$ and all its derivatives converge a.s point-wise its state $f_{\mathrm{init}}$ at initialization (47). We deduce that in the $\lambda \to \infty$ limit, the neural network in the lazy regime is equivalent to an untrained model $f_{\mathrm{init}}$, in terms of test error and robustness. This does not come as much of a surprise, since $\lambda \to \infty$, corresponds to early-stopping at $t = 0$, i.e., no optimization.

**Unregularized case $\lambda \to 0^+$.** By an analogous argument as above, $P_\lambda$ converges a.s. to the all-zero $m \times m$ matrix in the limit $\lambda \to 0^+$, and so thanks to (91), we have the almost-sure convergence $\|z_{\mathrm{RFL},\lambda} - z_{\mathrm{RF},\lambda}\| \to 0$. We deduce that in this limit, the unregularized lazy training regime is exactly equivalent to the unregularized vanilla RF regime. Thus, the random features lazy (RFL) regime corresponding to the approximation $f_{\mathrm{RFL}}$ is an interpolation between the random features regime (corresponding to $f_{\mathrm{RF}}$) and the untrained regime (corresponding to $f_{\mathrm{init}}$).

Although this is not useful in our infinite data regime, we remark that a non-zero amount of regularization is often crucial for good statistical performance with finite samples. In this, case, $P_\lambda$ is non-zero, and we expect both the test error and robustness to become worse in this lazy RF approximation, compared to vanilla RF.

### F.2  EFFECT OF REGULARIZATION IN RF REGIME

Suppose the estimation of the output weights of the RF model is regularized, i.e., for a fixed $\lambda \geq 0$, consider instead the model $f_{\mathrm{RF},\lambda}(x) := z_{\mathrm{RF},\lambda}^\top \sigma(Wx)$, where $z_{\mathrm{RF},\lambda}$ is chosen to solve the following

ridge-regularized problem

$$\min_{z \in \mathbb{R}^m} \|f_{W,z} - f_\star\|_{L^2(N(0,I_d))}^2 + \lambda\|z\|^2. \tag{52}$$

A simple computation gives the explicit form

$$z_{\text{RF},\lambda} = U_\lambda^{-1} v, \tag{53}$$

where $U_\lambda := U + \lambda I_m$, $U = U(W)$ is the random matrix defined in (15), and $v \in \mathbb{R}^m$ is random vector defined in (16). An inspection of the proof of Theorem 4.1 (see Appendix G.4) reveals that the situation in the presence of ridge regularization is equivalent to the unregularized case in which we replace $\bar\lambda$ by $\bar\lambda + \lambda$ in the definition of the matrix $A_0$ which appears in (20). This has the effect of decreasing $\psi_1$ and $\psi_2$, and thanks to (23), decreasing the robustness of the random features model. That is, $\widetilde\varepsilon_{\text{rob}}(f_{\text{RF},\lambda})$ is a decreasing function of the amount of regularization of $\lambda$, and in fact, $\lim_{\lambda \to \infty} \widetilde\varepsilon_{\text{rob}}(f_{\text{RF},\lambda}) = 0$.

# G TECHNICAL PROOFS

Before proving the main results of the manuscript, we first state and prove some auxiliary results which will be instrumental.

## G.1 A USEFUL LEMMA

Recall the definitions of the approximation error and robustness metrics from Section 2.3. The following lemma was used to express the measure of (non)robustness $\varepsilon_{\text{rob}}(f_{W,z,s})^2$ of a two-layer neural network $f_{W,z,s}$ as a quadratic form in the output weights, with coefficient matrix which depends on the distribution of the hidden weights.

Let us start by deriving an analytic formula for the robustness measure for the neural network general model (2). This result will be exploited in the sequel in the analysis of the different learning regimes we will consider.

**Lemma G.1.** *For the neural net $f_{W,z,s}$ defined in (2), we have the analytic formula $\varepsilon_{\text{rob}}(f_{W,z,s}) = z^\top C z$, where $C = C(W)$ is the $m \times m$ psd matrix with entries given by (with $x \sim N(0, I_d)$)*

$$c_{j,k} := (w_j^\top w_k)\mathbb{E}_x[\sigma'(x^\top w_j)\sigma'(x^\top w_k)] \, \forall j, k \in [m]. \tag{54}$$

*In particular, for a quadratic activation $\sigma(t) \equiv t^2 + s$, we have $c_{j,k} = 4(w_j^\top w_k)^2 \, \forall j, k \in [m]$.*

*Proof.* One directly computes $\nabla_x f_{W,z,s}(x) = \sum_{j=1}^m z_j \sigma'(x^\top w_j) w_j$, and so the Laplacian of $f_{W,z,s}$ at $x$ is given by

$$\|\nabla_x f_{W,z,s}(x)\|^2 = \sum_{j,k=1}^m z_j z_k (w_j^\top w_k)\sigma'(x^\top w_j)\sigma'(x^\top w_k). \tag{55}$$

Thus, $\mathfrak{S}(f_{W,z,s})^2$ evaluates to

$$\varepsilon_{\text{rob}}(f_{W,z,s}) := \mathbb{E}_{x \sim N(0,I_d)}\|\nabla_x f_{W,z,s}(x)\|^2 = \sum_{j,k=1}^m z_j z_k (w_j^\top w_k)\mathbb{E}_x[\sigma'(x^\top w_j)\sigma'(x^\top w_k)] = z^\top C(W)z,$$

where the $m \times m$ psd matrix $C(W)$ is as defined in Lemma G.1. In particular, for the activation function $\sigma(t) := t^2 + s$, one computes

$$c_{j,k} := (w_j^\top w_k)\mathbb{E}_{x \sim N(0,I_d)}[\sigma'(x^\top w_j)\sigma'(x^\top w_k)]$$
$$= 4(w_j^\top w_k)\mathbb{E}_{x \sim N(0,I_d)}[(x^\top w_j)(x^\top w_k)] = 4(w_j^\top w_k)^2,$$

where the last step is due to the fact that

$$\mathbb{E}_{x \sim N(0,I_d)}[(x^\top w_j)(x^\top w_k)] = \mathbb{E}_x[x^\top w_j w_k^\top x] = \text{tr}(\text{Cov}(x)w_j w_k^\top) = w_j^\top w_k,$$

by a standard result on the mean of a quadratic form. $\qquad\square$

**Corollary G.1** (Robustness error of teacher model). *It holds that $\varepsilon_{\text{rob}}(f_\star) = 4\|B\|_F^2$.*

*Proof.* For the first part follows directly from Lemma G.1 with activation function $\sigma(t) := t^2 + b_0/d$ and fixed output weight vector $z = 1_m := (1, \ldots, 1)$. $\qquad\square$

### G.2 HERMITE COEFFICIENTS

For any nonnegative integer $k$, let $\text{He}_k : \mathbb{R} \to \mathbb{R}$ be the (probabilist's) $k$th Hermite polynomial. For example, note that $\text{He}_0(t) := 0$, $\text{He}_1(t) := t$, $\text{He}_2(t) \equiv t^2 - 1$, $\text{He}_3(t) := t^3 - 3t$, etc. The sequence $(\text{He}_k)_k$ forms an orthonormal basis for the Hilbert space $L^2 = L^2(N(0,1))$ for functions $\mathbb{R} \to \mathbb{R}$ which are square-integrable w.r.t the standard normal distribution $N(0,1)$. Under suitable integrability conditions (refer to Section 4.1), the coefficients of the activation function $\sigma$ in this basis are called its *Hermite coefficients*, denoted $\lambda_k$, and are given by

$$\lambda_k = \lambda_k(\sigma) := \mathbb{E}_{G \sim N(0,1)}[\sigma(G)\text{He}_k(G)]. \tag{56}$$

Finally, $\|\sigma\|_{L^2(N(0,1))}^2 = \mathbb{E}_{G \sim N(0,1)}[\sigma(G)^2]$ defines the squared $L_2$-norm of $\sigma$ w.r.t the standard Gaussian distribution $N(0,1)$. Note that by construction, one has $\|\sigma\|_{L^2(N(0,1))}^2 = \sum_{k=0}^{\infty} \lambda_k^2(\sigma)$.

### G.3 APPROXIMATION OF RANDOM MATRICES

This section establishes some technical results for "linearizing" a number of complicated random matrices which occur in our analysis. We will make heavy use of random matrix theory (RMT) techniques developed in Silverstein & Choi (1995); El Karoui (2010); Ledoit & Péché (2011); Dobriban & Wager (2018)

We begin by recalling the following definition for future reference.

**Definition 4.1.** *With $z \sim N(0,1)$, define the following scalars*

$$\overline{\lambda} := \mathbb{E}[\sigma(z)^2] - \lambda_1^2, \ \kappa := \lambda_2^2 \|\Gamma\|_F^2 d/2, \ \tau := \lambda_2 \text{tr}(B\Gamma)\sqrt{d},$$
$$\overline{\lambda'} := \mathbb{E}[\sigma'(z)^2] - \lambda_1^2, \ \kappa' := \lambda_3^2 \|\Gamma\|_F^2 d/2. \tag{18}$$

Let $U$ be the random $m \times m$ psd matrix defined in (15) and let $v \in \mathbb{R}^m$ be the random vector defined in (16). Recall that $\lambda_k = \lambda_k(\sigma)$ is the $k$ Hermite coefficient of the activation function $\sigma$. Also recall the definition of the scalars $\overline{\lambda}, \kappa, \tau, \overline{\lambda'}$, and $\kappa'$ from (18). The following result was established in Ghorbani et al. (2019).

**Proposition G.1** (Lemma 2 of Ghorbani et al. (2019))**.** *If $\lambda_0 = 0$ and Conditions 4.1, 4.2 are in place, then in the limit (3), it holds that*

$$\|U - U_0\|_{op} = o_{d,\mathbb{P}}(1), \tag{57}$$

$$\|v - (\tau/\sqrt{d})1_m\| = o_{d,\mathbb{P}}(1), \tag{58}$$

*where the random $m \times m$ psd matrix $U_0$ is defined by*

$$U_0 := \overline{\lambda}I_m + \lambda_1^2 WW^\top + (\kappa/d)1_m 1_m^\top + \mu\mu^\top, \tag{59}$$

*and $\mu = (\mu_1, \ldots, \mu_m) \in \mathbb{R}^m$ with $\mu_i := \lambda_2 \cdot (\|w_i\|^2 - 1)/2$.*

A careful inspection of the proof of the estimate (57) reveals that we can remove the condition $\lambda_0 = 0$, at the expense of incurring rank-1 perturbations in the matrix $U_0$. Indeed, let us rewrite $\sigma = \overline{\sigma} + \lambda_0$, and with $\lambda_0(G) = \mathbb{E}_G[\overline{\sigma}(G)] = 0$ with $G \sim N(0,1)$ independent of the $w_i$'s. Let $T_0$ be the $m \times m$ matrix with entries $(T_0)_{ij} := \lambda_0(\sigma_i)\lambda_0(\sigma_j)$, where $\sigma_i$ is the function defined by $\sigma_i(z) := \sigma(\|w_i\|z) = \overline{\sigma}_i(z) + \lambda_0$, with $\overline{\sigma}(\|w_i\|z) := \overline{\sigma}_i(\|w_i\|z)$. Thus, we have the decomposition

$$T_0 = \overline{T}_0 + \lambda_0(u1_m^\top + 1_m u^\top) + \lambda_0^2 1_m 1_m^\top, \tag{60}$$

where $u = (\lambda_0(\overline{\sigma}_i))_{i \in [m]}$. Let $\overline{T}_0$ be the $m \times m$ psd matrix with entries $(\overline{T}_0)_{ij} := \lambda_0(\overline{\sigma}_i)\lambda_0(\overline{\sigma}_j)$. Using the arguments from Ghorbani et al. (2019) (since $\lambda_0(\overline{\sigma}) = 0$), one has

$$\|\overline{T}_0 - \mu\mu^\top\|_{op} = o_{d,\mathbb{P}}(1). \tag{61}$$

Furthermore, observe that one can write $u1_m^\top = R\mu1_m^\top$, where $R$ is the $m \times m$ diagonal matrix with $R_{ii} := \lambda_0(\overline{\sigma}_i)/\mu_i$. Now, for large $d$ and any $i \in [m]$, one computes

$$R_{ii} = \mathbb{E}_G\left[\frac{\sigma(\|w_i\|G) - \sigma(G)}{\lambda_2 \cdot (\|w_i\|^2 - 1)/2}\right] = \mathbb{E}_G\left[\frac{\sigma(\|w_i\|G) - \sigma(G)}{\|w_i\| - 1}\right] \cdot \frac{1}{\lambda_2 \cdot (\|w_i\| + 1)/2}$$
$$\to \frac{\mathbb{E}_G[G\sigma'(G)]}{\lambda_2 \cdot 2/2} = \frac{\lambda_2}{\lambda_2} = 1. \tag{62}$$

We deduce that $\|R - I_m\|_{op} = o_{d,\mathbb{P}}(1)$, and so $\|u1_m^\top - \mu 1_m^\top\|_{op} = o_{d,\mathbb{P}}(1)$. This proves the following extension of the above lemma which will be crucial in the sequel.

**Lemma G.2** (Linearization of $U$ without the Condition $\lambda_0(\sigma) = 0$). *Suppose Conditions 4.1 and 4.2 are in place. In the limit* (3)*, it holds that*

$$\|U - \widetilde{U}_0\|_{op} = o_{d,\mathbb{P}}(1), \tag{63}$$

*where $\widetilde{U}_0$ is the $m \times m$ random psd matrix given by*

$$\widetilde{U}_0 := \widetilde{\lambda} I_m + \lambda_1^2 WW^\top + (\kappa/d)1_m 1_m^\top + \widetilde{\mu}\widetilde{\mu}^\top, \tag{64}$$

*with $\widetilde{\mu} := \mu + \lambda_0 1_m$ and $\widetilde{\lambda} := \overline{\lambda} - \lambda_0^2 = \mathbb{E}_{G \sim N(0,1)}[\sigma(G)^2] - \lambda_0^2 - \lambda_1^2$.*

Let $C = C(W)$ be the random $m \times m$ psd matrix with entries given by

$$c_{ij} := (w_i^\top w_j)\mathbb{E}_{x \sim N(0,I_d)}[\sigma'(x^\top w_i)\sigma'(x^\top w_j)]. \tag{65}$$

Thanks to Lemma G.1, we know that $\varepsilon_{\mathrm{rob}}(f_{\mathrm{RF}}) = z_{\mathrm{RF}}^\top C z_{\mathrm{RF}} = v^\top U^{-1} C U^{-1} v$, a random quadratic form in $v$. We start by linearizing the nonlinear random coefficient matrix $C$.

**Lemma G.3** (Linearization of $C$). *Suppose Conditions 4.1 and 4.2 are in place. Then, in the limit* (3)*, we have the following approximation*

$$\|C - C_0\|_{op} = o_{d,\mathbb{P}}(1), \tag{66}$$

*where $C_0$ is the $m \times m$ random psd matrix given by*

$$C_0 := \overline{\lambda'} I_m + (\kappa'/d + \lambda_1^2)WW^\top + (2\kappa/d)1_m 1_m^\top, \tag{67}$$

*with $\kappa' := d \cdot \lambda_3^2 \|\Gamma\|_F^2/2 \geq 0$, and $\overline{\lambda'} := \|\sigma'\|_{L^2(N(0,1))}^2 - \lambda_1^2$.*

*Proof.* Note that $C = (WW^\top) \odot U'$, where $U'$ is the $m \times m$ random psd matrix with entries given by $U'_{ij} := \mathbb{E}_{x \sim N(0,I_m)}[\sigma'(x^\top w_i)\sigma'(x^\top w_j)]$.

– *Step 1: Linearization.* Invoking the previous lemma with $\sigma'$ in place of $\sigma$, we know that

$$\|U' - U'_0\|_{op} = o_{d,\mathbb{P}}(1), \tag{68}$$

where $U'_0$ is the $m \times m$ random matrix given by

$$
\begin{aligned}
U'_0 &:= \lambda' I_m + \lambda_1(\sigma')^2 WW^\top + (\kappa(\sigma')/d)1_m 1_m^\top + (\mu + \lambda_0(\sigma')1_m)(\mu + \lambda_0(\sigma')1_m)^\top \\
&= \lambda' I_m + \lambda_2(\sigma)^2 WW^\top + (\kappa'/d)1_m 1_m^\top + (\mu + \lambda_1(\sigma)1_m)(\mu + \lambda_1(\sigma)1_m)^\top,
\end{aligned} \tag{69}
$$

and we have used the fact that

$$\lambda_0((\sigma')^2) - \lambda_0(\sigma')^2 - \lambda_1(\sigma')^2 = \lambda_0((\sigma')^2) - \lambda_1(\sigma)^2 - \lambda_2(\sigma)^2 = \overline{\lambda'} - \lambda_2(\sigma)^2 =: \overline{\lambda'}.$$

Now, since $\|WW^\top\|_{op} = \mathcal{O}_{d,\mathbb{P}}(1)$ by standard RMT, we deduce that from (68) that,

$$
\begin{aligned}
\|C - (WW^\top) \odot U'\|_{op} &= \|(WW^\top) \odot (U' - U'_0)\|_{op} \\
&\leq \|WW^\top\|_{op} \cdot \|U' - U'_0\|_{op} \\
&= o_{d,\mathbb{P}}(1).
\end{aligned} \tag{70}
$$

– *Step 2: Simplification.* Let $E := \mathrm{diag}((\|w_i\|^2)_{i \in [m]})$ and $F := (WW^\top) \odot (WW^\top)$. Then

$$
\begin{aligned}
(WW^\top) \odot U'_0 &= \lambda' E + \lambda_2(\sigma)^2 F + (\kappa'/d)WW^\top + 2\lambda_1(\sigma)\mathrm{diag}(\mu)WW^\top + \lambda_1(\sigma)^2 WW^\top \\
&= \lambda' E + \lambda_1(\sigma)^2 F + (\kappa'/d + \lambda_1(\sigma)^2)WW^\top + 2\lambda_1(\sigma)\mathrm{diag}(\mu)WW^\top.
\end{aligned} \tag{71}
$$

Further, because $\max_{i \in [n]} |\|w_i\|^2 - 1| = o_{d,\mathbb{P}}(1)$ by basic concentration, we have

$$\|E - I_m\|_{op}, \|\mathrm{diag}(\mu)\|_{op} = o_{d,\mathbb{P}}(1). \tag{72}$$

Also, thanks to (El Karoui, 2010, Theorem 2.3), we may linearize $F$ like so

$$\|F - (I_m + \|\Gamma\|_F^2 1_m 1_m^\top)\|_{op} = o_{d,\mathbb{P}}(1).$$ (73)

Combining with (71) gives (recalling that $\kappa := d \cdot \lambda_2(\sigma)^2 \|\Gamma\|_F^2 / 2$)

$$
\begin{aligned}
(WW^\top) \odot U' &= (\lambda' + \lambda_2(\sigma)^2)I_m + (\kappa'/d + \lambda_1(\sigma)^2)WW^\top + (2\kappa/d)1_m 1_m^\top + \Delta \\
&= \overline{\lambda'}I_m + (\kappa'/d + \lambda_1(\sigma)^2)WW^\top + (2\kappa/d)1_m 1_m^\top + \Delta \\
&= C_0 + \Delta,
\end{aligned}
$$ (74)

where $\|\Delta\|_{op} = o_{d,\mathbb{P}}(1)$. □

Let us rewrite $U_0 = A_1 + \mu\mu^\top$ and $C_0 = D_0 + (2\kappa/d)1_m 1_m^\top$, where

$$
\begin{aligned}
A_1 &:= A_0 + (\kappa/d)1_m 1_m^\top, \\
A_0 &:= \widetilde{\lambda}I_m + \lambda_1^2 WW^\top, \\
D_0 &:= \overline{\lambda'}I_m + (\kappa'/d + \lambda_1^2)WW^\top.
\end{aligned}
$$ (75)

We will need the following lemmas.

**Lemma G.4.** *We have the following approximation*

$$
\begin{aligned}
\varepsilon_{\mathrm{rob}}(f_{\mathrm{RF}}) &= u^\top C_0 u + o_{d,\mathbb{P}}(1) \\
&= \tau^2 \frac{1_m^\top U_0^{-1} C_0 U_0^{-1} 1_m}{d} + o_{d,\mathbb{P}}(1),
\end{aligned}
$$ (76)

*where $u := U_0^{-1}h$, with $h := (\tau/\sqrt{d})1_m = \lambda_2 \cdot \mathrm{tr}(B\Gamma)1_m$ and $U_0$ is defined as in Proposition G.1 and $C_0$ is as defined in Lemma G.3.*

*Proof.* Thanks to Proposition G.1, the fitted output weights vector $z_{\mathrm{RF}} \in \mathbb{R}^m$ concentrates around $u := U_0^{-1}h$. On the other hand, we know from Lemma G.1 that $\varepsilon_{\mathrm{rob}}(f_{\mathrm{RF}}) = z_{\mathrm{RF}}^\top C z_{\mathrm{RF}}$. The result then follows from Lemma G.3. □

**Lemma G.5.** *Under Condition 4.3, the following holds in the limit (3)*

$$\frac{1_m^\top U_0^{-1} 1_m}{d} = \frac{\psi_1}{1 + \kappa\psi_1} + o_{d,\mathbb{P}}(1),$$ (77)

$$\frac{1_m^\top A_1^{-1} \mu}{\sqrt{d}} = o_{d,\mathbb{P}}(1),$$ (78)

$$\|A_1^{-1}\|_{op}, \|D_0\|_{op} = \mathcal{O}_{d,\mathbb{P}}(1).$$ (79)

*where $\psi_1 > 0$ is as defined in (20).*

*Proof.* Formula (77) was established in the proof of (Ghorbani et al., 2019, Theorem 1), whilst (78) was established in the proof of Lemma 5 of the same paper.

As for (79), we note that

$$\|A_1^{-1}\|_{op} = \|(\overline{\lambda}I_m + \lambda_1^2 WW^\top)^{-1}\|_{op} = \mathcal{O}_{d,\mathbb{P}}(1/\overline{\lambda}, \lambda_1^2) = \mathcal{O}_{d,\mathbb{P}}(1),$$

since $\overline{\lambda} = \Omega_d(1)$ under Condition 4.3. Similarly, one computes $\|D_0\|_{op} = \mathcal{O}_d(WW^\top) = \mathcal{O}_{d,\mathbb{P}}(1)$, by standard RMT arguments Vershynin (2012). □

We will need one final lemma.

**Lemma G.6.** *Let $A_1$, $A_0$, and $D_0$ be the random matrices defined in (75). Then, it holds that*

$$\frac{1_m^\top A_1^{-1} D_0 A_1^{-1} 1_m}{d} = \frac{\psi_2}{(1 + \kappa\psi_1)^2} + o_{d,\mathbb{P}}(1),$$ (80)

*where $\psi_1$ and $\psi_2$ as defined in (20).*

*Proof.* By Sherman-Morrison formula, we have

$$A_1^{-1} = A_0^{-1} - \kappa \frac{A_0^{-1} 1_m 1_m^\top A_0^{-1}/d}{(1 + \kappa 1_m^\top A_0^{-1} 1_m/d)},$$

and so $\frac{1_m^\top A_1^{-1} D_0 A_1^{-1} 1_m}{d} = a - 2ab - ab^2 = a(1-b)^2 = ac^2$, where

$$
\begin{aligned}
a &:= 1_m^\top A_0^{-1} D_0 A_0^{-1} 1_m/d, \\
b &:= \frac{\kappa 1_m^\top A_0^{-1} 1_m/d}{1 + \kappa 1_m^\top A_0^{-1} 1_m/d}, \\
c &:= 1 - b = \frac{1}{1 + \kappa 1_m^\top A_0^{-1} 1_m/d}.
\end{aligned}
\tag{81}
$$

Now, one has $1_m^\top A_0^{-1} 1_m/d = \operatorname{tr}(A_0^{-1})/d + o_{d,\mathbb{P}}(1)$, thanks to Lemmas 5 and 6 of Ghorbani et al. (2019). By an analogous argument, one can show that $1_m^\top A_0^{-1} D_0 A_0^{-1} 1_m/d = \operatorname{tr}(A_0^{-2} D_0)/d + o_{d,\mathbb{P}}(1)$. Finally, the fact that $\operatorname{tr}(A_0)^{-1}/d$ and $\operatorname{tr}(A_0^{-2} D_0)/d$ converge to deterministic values $\psi_1$ and $\psi_2$ respectively, can be established via standard RMT arguments Silverstein & Choi (1995); Ledoit & Péché (2011). □

### G.4 PROOF OF THEOREM 4.1: ANALYTIC FORMULA FOR ROBUSTNESS OF RF MODEL

We are now ready to prove Theorem 4.1, restated here for convenience.

**Theorem 4.1.** *Consider the random features model $f_{\mathrm{RF}}$ (13), with covariance matrix $\Gamma$ satisfying Condition 4.1 and activation function $\sigma$ satisfying Conditions 4.2 and 4.3.*

*(A) In the limit (3), we have the following approximation*

$$\widetilde{\varepsilon}_{\mathrm{rob}}(f_{\mathrm{RF}}) = \frac{\tau^2 (2\kappa \psi_1^2 + \psi_2)}{\|B\|_F^2 (2\kappa \psi_1 + 2)^2} + o_{d,\mathbb{P}}(1). \tag{23}$$

*(B) Moreover, if $\lim_{d\to\infty} \alpha = \alpha_\infty$, then $\lim_{\rho\to\infty} \lim_{\substack{m,d\to\infty \\ m/d\to\rho}} \widetilde{\varepsilon}_{\mathrm{rob}}(f_{\mathrm{RF}}) = \alpha_\infty^2$ w.p 1. In particular, for the optimal choice of $\Gamma$ in terms of test error, namely $\Gamma \propto B$, one has $\lim_{\rho\to\infty} \lim_{\substack{m,d\to\infty \\ m/d\to\rho}} \widetilde{\varepsilon}_{\mathrm{rob}}(f_{\mathrm{RF}}) = 1$ w.p 1.*

*Proof.* From Lemmas G.1 and G.3, we know that

$$\varepsilon_{\mathrm{rob}}(f_{\mathrm{RF}}) = z_{\mathrm{RF}}^\top C z_{\mathrm{RF}} = u^\top C_0 u + o_{d,\mathbb{P}}(1) = \tau^2 \frac{1_m^\top U_0^{-1} C_0 U_0^{-1} 1_m}{d} + o_{d,\mathbb{P}}(1), \tag{82}$$

where $u := U_0^{-1} h$, with $h := (\tau/\sqrt{d}) 1_m = \lambda_2 \cdot \operatorname{tr}(B\Gamma) 1_m$ and $U_0$ defined as in Lemma G.2 and $C$, $C_0$ are as defined in Lemma G.3. Let $A_1$, $A_0$, and $D_0$ be the random matrices defined in (75). Since $C_0 = D_0 + (2\kappa/d) 1_m 1_m^\top$, one computes

$$
\begin{aligned}
\frac{1_m^\top U_0^{-1} C_0 U_0^{-1} 1_m}{d} &= \frac{1_m^\top U_0^{-1} D_0 U_0^{-1} 1_m}{d} + 2\kappa \cdot \frac{1_m^\top U_0^{-1} 1_m 1_m^\top U_0^{-1} 1_m}{d^2} \\
&= \frac{1_m^\top U_0^{-1} D_0 U_0^{-1} 1_m}{d} + 2\kappa \cdot \left( \frac{1_m^\top U_0^{-1} 1_m}{d} \right)^2 \\
&= \frac{1_m^\top U_0^{-1} D_0 U_0^{-1} 1_m}{d} + \frac{2\kappa \psi_1^2}{(1 + \kappa \psi_1)^2} + o_{d,\mathbb{P}}(1),
\end{aligned}
\tag{83}
$$

where the last step is thanks to Lemma G.5. It remains to estimate the first term in the above display.

Using the Sherman-Morrison formula, we have

$$U_0^{-1} = A_1^{-1} - \frac{A_1^{-1} \mu \mu^\top A_1^{-1}}{1 + \mu^\top A_1^{-1} \mu}. \tag{84}$$

We deduce that

$$\frac{1_m^\top U_0^{-1} D_0 U_0^{-1} 1_m}{d} = a_{11} - a_{12} - a_{21} + a_{22} + o_{d,\mathbb{P}}(1), \tag{85}$$

where $a_{11}$, $a_{12}$, $a_{21}$, and $a_{22}$ are defined by

$$
\begin{aligned}
a_{11} &:= \frac{1_m^\top A_1^{-1} D_0 A_1^{-1} 1_m}{d}, \\
a_{12} &:= \frac{1_m^\top A_1^{-1} D_0 A_1^{-1} \mu \mu^\top A_1^{-1} 1_m}{(1 + \mu^\top A_1^{-1} \mu) d}, \\
a_{21} &:= \frac{1_m^\top A_1^{-1} D_0 A_1^{-1} \mu \mu^\top A_1^{-1} 1_m}{(1 + \mu^\top A_1^{-1} \mu) d}, \\
a_{22} &:= \frac{1_m^\top A_1^{-1} \mu \mu^\top A_1^{-1} D_0 A_1^{-1} \mu \mu^\top A_1^{-1} 1_m}{(1 + \mu^\top A_1^{-1} \mu)^2 d}.
\end{aligned}
\tag{86}
$$

Now, one easily computes

$$\max(|a_{12}|, |a_{21}|) \le \|D_0\|_{op} \|A_1^{-1}\|_{op} \cdot \frac{1_m^\top A_1^{-1} \mu \mu^\top A_1^{-1} 1_m}{(1 + \mu^\top A_1^{-1} \mu) d} \lesssim \frac{(1_m^\top A_1^{-1} \mu / \sqrt{d})^2}{(1 + \mu^\top A_1^{-1} \mu)} = o_{d,\mathbb{P}}(1),$$

where we have used Lemma G.5 in the last two steps. Similarly, we have,

$$|a_{22}| \le \underbrace{\|D_0\|_{op} \|A_1^{-1}\|_{op}}_{\mathcal{O}_{d,\mathbb{P}}(1)} \cdot \underbrace{1_m^\top A_1^{-1} \mu / \sqrt{d}}_{o_{d,\mathbb{P}}(1)} \cdot \underbrace{\frac{\mu^\top A_1^{-1} \mu}{(1 + \mu^\top A_1^{-1} \mu)^2}}_{\mathcal{O}_{d,\mathbb{P}}(1)} \cdot \underbrace{\mu^\top A_1^{-1} 1_m / \sqrt{d}}_{o_{d,\mathbb{P}}(1)} = o_{d,\mathbb{P}}(1),$$

again thanks to Lemma G.5. We conclude from (85) that

$$\frac{1_m^\top U_0^{-1} D_0 U_0^{-1} 1_m}{d} = a_{11} + o_{d,\mathbb{P}}(1). \tag{87}$$

Finally, we know from Lemma G.6 that

$$a_{11} := \frac{1_m^\top A_1^{-1} D_0 A_1^{-1} 1_m}{d} = \frac{\psi_2}{(1 + \kappa \psi_1)^2} + o_{d,\mathbb{P}}(1).$$

part (A) of the theorem them follows upon dividing (85) by $\varepsilon_{\text{rob}}(f_\star) = 4\|B\|_F^2$.

For part (B), one notes that $\psi_1 > 0$ and so

$$
\begin{aligned}
\frac{\tau^2 (2\kappa \psi_1^2 + \psi_2)}{\|B\|_F^2 (2\kappa \psi_1 + 2)^2} &= \frac{\text{tr}(B\Gamma)^2 d(\|\Gamma\|_F^2 d \psi_1 + \psi_2)}{(\|\Gamma\|_F^2 d \psi_1 + 2)^2 \|B\|_F^2} = \frac{\text{tr}(B\Gamma)^2 d^2 \|\Gamma\|_F^2 d \psi_1}{(\|\Gamma\|_F^2 d \psi_1 + 2)^2 \|B\|_F^2} + o_d(1) \\
&= \frac{\text{tr}(B\Gamma)^2}{\|\Gamma\|_F^2 \|B\|_F^2} + o_d(1) \to \alpha_\infty^2,
\end{aligned}
$$

which completes the proof. $\qquad\qquad\square$

# H  PROOFS OF MAIN RESULTS

## H.1  PROOF OF THEOREM E.1: ROBUSTNESS ERROR OF NEURAL NETWORKS AT INITIALIZATION

We restate the result here for convenience. Let $f_{\text{init}}$ be the function computed by the neural network at initialization, as defined in (47).

**Theorem E.1.** *Under the Conditions 4.1 and 4.2, we have the identity in the limit* (3),

$$\widetilde{\varepsilon}_{\text{rob}}(f_{\text{init}}) = \frac{\|\sigma'\|_{L^2(N(0,1))}^2 + \lambda_3^2 \|\Gamma\|_F^2 / 2 + \lambda_2^2 \|\Gamma\|_F^2}{4\|B\|_F^2} + o_{d,\mathbb{P}}(1), \text{ where } \lambda_k \text{ is the kth Hermite coef-}$$

*ficient of the activation function $\sigma$. In particular, for the quadratic activation function $\sigma(t) = t^2 - 1$, we have* $\widetilde{\varepsilon}_{\text{rob}}(f_{\text{init}}) = \frac{1 + \|\Gamma\|_F^2}{\|B\|_F^2} + o_{d,\mathbb{P}}(1)$.

*Proof.* Thanks to Lemma G.1, we know that $\varepsilon_{\mathrm{rob}}(f_{\mathrm{init}}) = z^\top C z$, where $C$ is the random $m \times m$ psd matrix defined in (65). By standard RMT, $z^\top C z = \mathrm{tr}(C)/m + o_{d,\mathbb{P}}(1)$. Now, let $C_0$ be the random matrix introduced in Lemma G.3. Since $\|C - C_0\|_{op} = o_{d,\mathbb{P}}(1)$ (thanks to the aforementioned lemma), one has $\mathrm{tr}(C)/m = \mathrm{tr}(C_0)/m + o_{d,\mathbb{P}}(1)$. Let $D_0 := \overline{\lambda'}I_m + (\kappa'/d + \lambda_1^2)WW^\top$ be the matrix defined in (75) so that $C_0 = D_0 + (2\kappa/d)1_m 1_m^\top$. We deduce that in the limit (3),

$$
\begin{aligned}
\varepsilon_{\mathrm{rob}}(f_{\mathrm{init}}) &= \mathrm{tr}(D_0)/m + 2\kappa/d + o_{d,\mathbb{P}}(1) \\
&= (\kappa'/d + \lambda_1^2)\mathrm{tr}(WW^\top)/m + \overline{\lambda'} + 2\kappa/d + o_{d,\mathbb{P}}(1) \\
&= k'/d + \lambda_1^2 + \overline{\lambda'} + 2\kappa/d + o_{d,\mathbb{P}}(1) \\
&= \|\sigma'\|_{L^2(N(0,1))}^2 + \kappa'/d + 2\kappa/d + o_{d,\mathbb{P}}(1) \\
&= \|\sigma'\|_{L^2(N(0,1))}^2 + \lambda_3^2\|\Gamma\|_F^2/2 + \lambda_2^2\|\Gamma\|_F^2 + o_{d,\mathbb{P}}(1)
\end{aligned}
\tag{88}
$$

where the third line is because $\mathrm{tr}(WW^\top)/m = (1/m)\sum_{j=1}^m \|w_j\|^2$ which converges in probability to $\mathrm{tr}(\Gamma) = 1$, by the weak law of large numbers. Dividing by both sides of the above display by $\varepsilon_{\mathrm{rob}}(f_\star) = 4\|B\|_F^2$ then gives the result.

In particular, in the case of quadratic activation $\sigma(t) := t^2 - 1$, we have $\lambda_2 = 2$, $\|\sigma'\|_{L^2(N(0,1))}^2 = 4$, $\lambda_3 = 0$, and so we deduce that $\widetilde{\varepsilon}_{\mathrm{rob}}(f_{\mathrm{init}}) = (4 + 4\|\Gamma\|_F^2)/(4\|B\|_F^2) = (1 + \|\Gamma\|_F^2)/\|B\|_F^2$. $\square$

## H.2 PROOF OF THEOREM E.2: TEST ERROR OF NEURAL NETWORK AT INITIALIZATION

**Theorem E.2.** *Under the Conditions 4.1 and 4.2, we have the following identity in the limit (3),*
$$
\widetilde{\varepsilon}_{\mathrm{test}}(f_{\mathrm{init}}) = 1 + \frac{\|\sigma\|_{L^2(N(0,1))}^2 + \lambda_2^2\|\Gamma\|_F^2/2}{2\|B\|_F^2} + o_{d,\mathbb{P}}(1). \text{ In particular, for the quadratic activation}
$$
$\sigma(t) := t^2 - 1$, *we have the following identity* $\widetilde{\varepsilon}_{\mathrm{test}}(f_{\mathrm{init}}) = 1 + \dfrac{1 + \|\Gamma\|_F^2}{\|B\|_F^2} + o_{d,\mathbb{P}}(1)$.

*Proof.* For random initial output weights $z^0 \sim N(0, (1/m)1_m)$ independent of the (random) hidden weights matrix $W$, one computes

$$
\mathbb{E}_z[\varepsilon_{\mathrm{test}}(f_{\mathrm{init}})] := \mathbb{E}_z\mathbb{E}_{x \sim N(0,I_d)}[(f_{\mathrm{init}}(x) - f_\star(x))^2] = \mathbb{E}_z\mathbb{E}_x[f_{\mathrm{init}}(x)^2] + \mathbb{E}_x[f_\star(x)^2], \tag{89}
$$

where we have used the fact that $\mathbb{E}z = 0$. The second term in the rightmost expression equals $\|f_\star\|_{L^2(N(0,I_d))}^2 = 2\|B\|_F^2$. Let $Q$ be the $m \times m$ diagonal matrix with the output weights $z$ on the diagonal, and let $U$ be the $m \times m$ matrix with entries $U_{ij} := \mathbb{E}_x[\sigma(x^\top w_j)\sigma(x^\top w_j)]$ introduced in (15), and let $U_0 := \overline{\lambda}I_m + \lambda_1^2 WW^\top + (\kappa/d)1_m 1_m^\top + \mu\mu^\top$ with $\mu := (\lambda_2(\|w_j\|^2 - 1))_{j \in [m]} \in \mathbb{R}^m$, be its approximation given in Proposition G.1. Then

$$
\begin{aligned}
\mathbb{E}_x[f_{\mathrm{init}}(x)^2] &= \mathbb{E}_x[\sigma(Wx)^\top Q\sigma(Wx)] = z^\top \mathbb{E}_x[\sigma(Wx)\sigma(Wx)^\top]z = z^\top U z \\
&= \mathrm{tr}(U)/m + o_{d,\mathbb{P}}(1), \text{ by concentration of random quadratic forms} \\
&= \mathrm{tr}(U_0)/m + o_{d,\mathbb{P}}(1), \text{ thanks to Proposition G.1} \\
&= \overline{\lambda} + \lambda_1^2 \underbrace{\mathrm{tr}(WW^\top)/m}_{1 + o_{d,\mathbb{P}}(1)} + k/d + \lambda_2 \underbrace{\sum_{i=1}^m (\|w_i\|^2 - 1)^2/m}_{o_{d,\mathbb{P}}(1)} + o_{d,\mathbb{P}}(1) \\
&= \overline{\lambda} + \lambda_1^2 + \kappa/d + o_{d,\mathbb{P}}(1) \\
&= \|\sigma\|_{L^2(N(0,1))}^2 + \lambda_2^2\|\Gamma\|_F^2/2 + o_{d,\mathbb{P}}(1).
\end{aligned}
\tag{90}
$$

The first part of the result then follows upon dividing through by $\|f_\star\|_{L^2(N(0,I_d))}^2 = 2\|B\|_F^2$.

In particular, if $\sigma$ is the quadratic activation, then $\|\sigma\|_{L^2(N(0,I_d))}^2 = \lambda_2 = 2$, and the second part of the result follows. $\square$

### H.3 THE SPECIAL CASE OF QUADRATIC ACTIVATIONS

We now specialize Theorem 4.1 to the case of the quadratic activation function and obtain more transparent formulae.

**Corollary H.1.** *Consider the random features model $f_{\mathrm{RF}}$ with covariance matrix $\Gamma$ satisfying Condition 4.1 and quadratic activation function $\sigma(t) := t^2 - 1$. Then, in the limit (3), it holds that*

$$\widetilde{\varepsilon}_{\mathrm{rob}}(f_{\mathrm{RF}}) = \frac{\mathrm{tr}(B\Gamma)^2 \|\Gamma\|_F^2}{(1/m + \|\Gamma\|_F^2)^2 \|B\|_F^2} + o_{d,\mathbb{P}}(1).$$ *Furthermore, part (B) of Theorem 4.1 holds.*

*Proof.* For quadratic activation, one easily computes

$$\lambda_1 = \lambda_0 = 0, \ \lambda_2 = 2, \ \overline{\lambda} = 2, \ \overline{\lambda'} = 4, \ \kappa = \lambda_2^2 \|\Gamma\|_F^2 d/2 = 2\|\Gamma\|_F^2 d, \ \tau := 2\mathrm{tr}(B\Gamma)/\sqrt{d}, \ \kappa' = 0,$$

and so $A_0 = 2I_m$ and $D_0 = 4I_m$. In this case, one deduces

$$\psi_1 := \lim_{\substack{m,d\to\infty \\ d/m\to\rho}} \mathrm{tr}(A_0^{-1})/d = \rho/2 \quad \text{and} \quad \psi_2 := \lim_{\substack{m,d\to\infty \\ d/m\to\gamma}} \mathrm{tr}(A_0^{-2}D_0)/d = \rho.$$

Plugging these into formula (23) of Theorem 4.1 yields

$$\begin{aligned}
\widetilde{\varepsilon}_{\mathrm{rob}}(f_{\mathrm{RF}}) &= \frac{4\mathrm{tr}(B\Gamma)^2 d \cdot 2 \cdot 2\|\Gamma\|_F^2 d \cdot (\rho/2)^2}{(2 + 2 \cdot 2\|\Gamma\|_F^2 d \cdot \rho/2)^2 \|B\|_F^2} + o_{d,\mathbb{P}}(1) \\
&= \frac{4\mathrm{tr}(B\Gamma)^2 \|\Gamma\|_F^2 (\rho d)^2}{(2 + 2\|\Gamma\|_F^2 \rho d)^2 \|B\|_F^2} + o_{d,\mathbb{P}}(1) \\
&= \frac{\mathrm{tr}(B\Gamma)^2 \|\Gamma\|_F^2}{(1/m + \|\Gamma\|_F^2)^2 \|B\|_F^2} + o_{d,\mathbb{P}}(1),
\end{aligned}$$

which proves the first part of the corollary. The second part follows directly from the second part of Theorem 4.1. $\qquad\square$

### H.4 PROOF OF THEOREM F.1: RANDOM FEATURES LAZY (RFL) REGIME

**Theorem F.1.** *We have the following identities*

$$\mathbb{E}_{z^0}[\widetilde{\varepsilon}_{\mathrm{test}}(f_{\mathrm{RFL},\lambda})] = \widetilde{\varepsilon}_{\mathrm{test}}(f_{\mathrm{RF}}) + \frac{\mathrm{tr}(P_\lambda^2 U)/m}{2\|B\|_F^2} + o_{d,\mathbb{P}}(1) \tag{50}$$

$$\mathbb{E}_{z^0}[\widetilde{\varepsilon}_{\mathrm{rob}}(f_{\mathrm{RFL},\lambda})] = \widetilde{\varepsilon}_{\mathrm{rob}}(f_{\mathrm{RF}}) + \frac{\mathrm{tr}(P_\lambda^2 C)/m}{4\|B\|_F^2} + o_{d,\mathbb{P}}(1), \tag{51}$$

*where $U = U(W)$ and $C = C(W)$ are the random matrices defined in (15) and (54) respectively.*

*Proof.* By construction, note that the vector $\delta_\lambda$ is equivalent to the output weights of a RF approximation with true labels $\widetilde{f}_\star(x) := f_\star(x) - f_{\mathrm{init}}(x)$. If $U$ and $v$ are as defined in (15) and (16) respectively, then we have the closed-form solution (with $U_\lambda := U + \lambda I_m$)

$$\begin{aligned}
\delta_\lambda &= U_\lambda^{-1}(\mathbb{E}_x[(f_\star(x) - f_{z^0}(x))\sigma(Wx)]) \\
&= U_\lambda^{-1}(v - \mathbb{E}_x[(z^0)^\top \sigma(Wx)\sigma(Wx)^\top]) \\
&= U_\lambda^{-1}(v - Uz^0) = z_{\mathrm{RF},\lambda} - U_\lambda^{-1}Uz^0.
\end{aligned}$$

Thus, for a fixed regularization parameter $\lambda > 0$, the output weights vector in this lazy training regime is given by

$$z_{\mathrm{RFL},\lambda} = \delta_\lambda + z^0 = z_{\mathrm{RF},\lambda} + P_\lambda z^0, \tag{91}$$

where $P_\lambda := I_m - U_\lambda^{-1}U$. We deduce that in the presence of any amount of ridge regularization, the lazy random features (RFL) regime is equivalent to the vanilla random features (RF) regime, with an additive bias of $P_\lambda z^0 \in \mathbb{R}^m$ on the fitted output weights vector. In particular, note that if $\lambda = 0$, then $z_{\mathrm{RFL},0} = z_{\mathrm{RF},0}$, that is in the absence of regularization, the RFL and RF correspond to the same regime (i.e., the initialization has no impact on the final model).

– *test error.* From formula (91), and noting that $z^0$ is independent of $W$, one computes the test error of $f_{\text{lazy},\lambda}$ averaged over the initial output weights vector $z^0$ as

$$
\begin{aligned}
\mathbb{E}_{z^0}[\varepsilon_{\text{test}}(f_{\text{lazy},\lambda})] &:= \mathbb{E}_{z^0}[\|f_{\text{lazy},\lambda} - f_\star\|^2_{L^2(N(0,I_d))}] \\
&= \|f_{\text{RF}} - f_\star\|^2_{L^2(N(0,I_d))} + \mathbb{E}_{z^0}[\|f_{W,P_\lambda z^0}\|^2_{L^2(N(0,I_d))}] \\
&= \varepsilon_{\text{test}}(f_{\text{RF},\lambda}) + \mathbb{E}_{a_0}[(z^0)^\top P_\lambda U P_\lambda z^0] \\
&= \varepsilon_{\text{test}}(f_{\text{RF},\lambda}) + \operatorname{tr}(P_\lambda^2 U)/m,
\end{aligned}
$$

where $U = U(W)$ is the matrix defined in (15).

– *(Non)robustness.* From formula (91), one computes

$$
\begin{aligned}
\mathfrak{S}(f_{\text{RFL}},\lambda)^2 = z_{\text{RFL},\lambda}^\top C z_{\text{RFL},\lambda} &= z_{\text{RF},\lambda}^\top C z_{\text{RF},\lambda} + 2 z_{\text{RF},\lambda} C P_\lambda z^0 + (z^0)^\top P_\lambda C P_\lambda z^0 \\
&= \mathfrak{S}(f_{\text{RF},\lambda})^2 + 2 z_{\text{RF},\lambda} C P_\lambda z^0 + (z^0)^\top P_\lambda C P_\lambda z^0,
\end{aligned}
$$

where $C = C(W)$ is the matrix defined in (65). Taking expectations w.r.t $z^0$, and noting that $z^0$ is independent of $P_\lambda$ and $C$ only depend on $W$ and are therefore independent of $z^0$, we have

$$
\mathbb{E}_{z^0}[\mathfrak{S}(f_{\text{lazy},\lambda})^2] = \mathfrak{S}(f_{\text{RF},\lambda})^2 + \operatorname{tr}(P_\lambda^2 C)/m. \tag{92}
$$

$\square$

## H.5 PROOF OF THEOREM 5.1: NEURAL TANGENT (NT) REGIME

**Theorem 5.1.** *Consider the neural tangent model $f_{\text{NT}}$ in (26). In the limit (3) it holds that,*

$$
\mathbb{E}_W[\tilde{\varepsilon}_{\text{rob}}(f_{\text{NT}})] = (\underline{\rho} + \underline{\rho}^2)/2 + (\underline{\rho} - \underline{\rho}^2)\beta/2 + o_d(1), \ \text{where } \underline{\rho} := \min(\rho, 1). \tag{28}
$$

Let $r \leq \min(m,d)$ be the rank of $W$. It is clear that $r = \min(m,d)$ w.p 1. Let

$$
W^\top = P_1 S V^\top \tag{93}
$$

be the singular-value decomposition of $W^\top$, where $P_1 \in \mathbb{R}^{d \times r}$ (resp. $V \in \mathbb{R}^{m \times r}$) is the column-orthogonal matrix of singular-vectors of $W^\top$ (resp. $W$), and $S \in \mathbb{R}^{r \times r}$ is the diagonal matrix of nonzero singular-values. For any $A \in \mathbb{R}^{m \times d}$, set $G(A) := SV^\top A \in \mathbb{R}^{r \times d}$. In their proof of (27), Ghorbani et al. (2019) showed that it is optimal (in terms of test error) to chose $A_{\text{NT}}$ such that $G(A_{\text{NT}}) = P_1^\top B/2$. Multiplying through by the orthogonal projection matrix $P_1$ gives

$$
P_1 P_1^\top B/2 = P_1 G(A_{\text{NT}}) = P_1 S V^\top A_{\text{NT}} = W^\top A_{\text{NT}}. \tag{94}
$$

For the proof of Theorem 5.1, we will need the following lemma.

**Lemma H.1.** $\varepsilon_{\text{rob}}(f_{\text{NT}}) = 4\|W^\top A + A^\top W\|_F^2$.

*Proof.* Note that we can rewrite

$$
f_{\text{NT}}(x) = 2\operatorname{tr}((W^\top A)xx^\top) - c,
$$

which is linear in $xx^\top \in \mathbb{R}^{d \times d}$. One then readily computes $\nabla f_{\text{NT}}(x) = 2(W^\top A + A^\top W)x$, from which we deduce that $\|\nabla f_{\text{NT}}(x)\|^2 = 4x^\top (W^\top A + A^\top W)^2 x$. Averaging over $x \sim N(0, I_d)$ then gives

$$
\begin{aligned}
\frac{\varepsilon_{\text{rob}}(f_{\text{NT}})}{4} &:= \mathbb{E}_x \|\nabla f_{\text{NT}}(x)\|^2 = \mathbb{E}_x[x^\top (W^\top A + A^\top W)^2 x] \\
&= \operatorname{tr}((W^\top A + A^\top W)^2) = \|W^\top A + A^\top W\|_F^2,
\end{aligned}
$$

which completes the proof. $\square$

We will also need the following auxiliary lemma.

**Lemma H.2.** *Let $P_1$ be as in* (93) *and let $\beta := \operatorname{tr}(B)^2/(d\|B\|_F^2)$ as usual. In the limit* (3)*, we have the identities*

$$\mathbb{E}_W \|P_1 P_1^\top B\|_F^2 = \|B\|_F^2(\underline{\rho} + o_d(1)), \tag{95}$$

$$\mathbb{E}_W \|P_1^\top B P_1\|_F^2 = \|B\|_F^2(\underline{\rho}^2(1-\beta) + \underline{\rho}\beta + o_d(1)), \tag{96}$$

*where $\underline{\rho} := \min(\rho, 1)$.*

*Proof.* WLOG, let $B$ be a diagonal matrix, so that $B^2 = \sum_j \lambda_j^2 e_j e_j^\top$, where $e_j$ is the $j$th standard unit-vector in $\mathbb{R}^d$. Then, with $P = P_1 P_1^\top$, we have

$$\begin{aligned}
\|P_1 P_1^\top B\|_F^2 = \operatorname{tr}(PB^2) &= \sum_i (PB^2)_{ii} = \sum_{i,j} P_{ij}(B^2)_{ji} \\
&= \sum_{i,j} P_{ij}(B^2)_{ji} = \sum_{i,j} \lambda_i P_{ij} \delta_{ij}^2 = \sum_j \lambda_j P_{jj}.
\end{aligned} \tag{97}$$

Therefore, $\mathbb{E}_W[\|P_1 P_1^\top B\|_F^2] = (1/d)\mathbb{E}_W[\operatorname{tr}(P)] \cdot \sum_j \lambda_j^2 = \min(m/d, 1)\|B\|_F^2 = \|B\|^2(\underline{\rho} + o_d(1))$, where we have used the fact that $\mathbb{E}_W P_{jj} = (1/d)\mathbb{E}_W \operatorname{tr}(P)$ for all $j$, due to rotation-invariance. This proves (95).

The proof of (96) is completely analogous to the proof of formula (69) in Ghorbani et al. (2019), with $\rho$ therein replaced with $1 - \underline{\rho}$, and is thus omitted. $\qquad\square$

*Proof of Theorem 5.1.* From Lemma H.1 and formula (94)), we know that

$$\begin{aligned}
\varepsilon_{\mathrm{rob}}(f_{\mathrm{NT}}) &= 4\|W^\top A_{\mathrm{NT}} + A_{\mathrm{NT}}^\top W\|_F^2 \\
&= 4\|P_1 P_1^\top B/2 + B P_1 P_1^\top/2\|_F^2 \\
&= 2\|P_1 P_1^\top B\|_F^2 + 2\|P_1^\top B P_1\|_F^2.
\end{aligned} \tag{98}$$

The result then follows upon taking expectations w.r.t the hidden weights matrix $W$ and applying Lemma H.2. $\qquad\square$

## H.6 Proof of Theorem 5.2: Neural tangent lazy (NTL) regime

**Theorem 5.2.** *Suppose the output weights $z^0$ at initialization are iid from $N(0, (1/m)I_m)$. Then, in the limit* (3)*, the following identities hold*

$$\mathbb{E}_{\{W,z^0\}}[\widetilde{\varepsilon}_{\mathrm{test}}(f_{\mathrm{NTL}})] = \mathbb{E}_W[\widetilde{\varepsilon}_{\mathrm{test}}(f_{\mathrm{NT}})] + o_d(1), \tag{31}$$

$$\mathbb{E}_{\{W,z^0\}}[\widetilde{\varepsilon}_{\mathrm{rob}}(f_{\mathrm{NTL}})] = \mathbb{E}_W[\widetilde{\varepsilon}_{\mathrm{rob}}(f_{\mathrm{NT}})] + \mathbb{E}_{\{W,z^0\}}[\widetilde{\varepsilon}_{\mathrm{rob}}(f_{\mathrm{init}})] + o_d(1). \tag{32}$$

*Proof.* First observe that $f_\star(x) - f_{\mathrm{NTL}}(x; A, c) = \widetilde{f}_\star(x) - f_{\mathrm{NT}}(x; A, c)$, where,

$$\widetilde{f}_\star(x) := f_\star(x) - f_{\mathrm{init}}(x) = x^\top \widetilde{B} x + b_0, \tag{99}$$

and the $d \times d$ matrix $\widetilde{B}$ is defined by

$$\widetilde{B} := B - W^\top Q W. \tag{100}$$

Thus, fitting the model $f_{\mathrm{NTL}}(\cdot; A, c)$ to the teacher model $f_\star$ with coefficient matrix $B$ is equivalent to fitting $f_{\mathrm{NT}}(\cdot; A, c)$ to the modified teacher model $\widetilde{f}_\star$ with coefficient matrix $\widetilde{B}$.

In terms of test error (4), let $A_{\mathrm{NTL}}, c_{\mathrm{NTL}}$ be optimal in $f_{\mathrm{NT}}(\cdot; A, c)$, and for simplicity of notation define

$$f_{\mathrm{NTL}}(x) := f_{\mathrm{NTL}}(x; A_{\mathrm{NTL}}, c_{\mathrm{NTL}}). \tag{101}$$

We split the proof into two parts. In the first part, we establish (32). The second part handles (31).

– *Robustness error.* Proceeding in the same way as in the paragraph leading to (94), one has

$$\varepsilon_{\rm rob}(f_{\rm NTL}) = 2\|P_1 P_1^\top \widetilde{B}\|_F^2 + 2\|P_1^\top \widetilde{B} P_1\|_F^2, \tag{102}$$

where $P_1 \in \mathbb{R}^{d \times r}$ is the column-orthogonal matrix in (93) and $r := \min(m, d)$ is the rank of $W$ (w.p 1). Now, by definition of $\widetilde{B}$, one has $\widetilde{B}^2 = (B - W^\top QW)(B - W^\top QW)$, and so

$$P_1 P_1^\top \widetilde{B}^2 = P_1 P_1^\top B^2 - P_1 P_1^\top B W^\top QW - P_1 P_1^\top W^\top QW B + P_1 P_1^\top W^\top QW W^\top QW. \tag{103}$$

We now take the expectation w.r.t $(W, z^0)$, of each term on the RHS. Thanks to Lemma H.2, we recognize the expectation w.r.t $W$ of the trace of the first term in (103) as

$$\mathbb{E}_W[\text{tr}(P_1 P_1^\top B^2)] = \mathbb{E}_W[\|P_1 P_1^\top B\|_F^2] = \|B\|_F^2 (\underline{\rho} + o_d(1)), \tag{104}$$

Now, since $W$ and $z^0$ are independent and $z^0$ has zero mean, the second and third terms in (102) have zero expectation w.r.t $(W, z^0)$ because they are linear in $Q = \text{diag}(z^0)$.

Finally, one notes that

$$P_1 P_1^\top W^\top QW W^\top QW = P_1 SV^\top D W W^\top QV S P_1^\top = W^\top QW W^\top QW, \tag{105}$$

and so taking expectation w.r.t $W$ and $D$ (i.e $z^0$) yields

$$\begin{aligned}
\mathbb{E}_{\{W,z^0\}}[\text{tr}(P_1 P_1^\top W^\top QW W^\top QW)] &= \mathbb{E}_{\{W,z^0\}}[\text{tr}(W W^\top QW W^\top Q)] \\
&= \mathbb{E}_{\{W,z^0\}}[z^\top((WW^\top) \odot (WW^\top))z] \\
&= \frac{1}{4}\mathbb{E}_{\{W,z^0\}}[\varepsilon_{\rm rob}(f_{\rm init})],
\end{aligned} \tag{106}$$

where the last step is thanks to the second part of Lemma G.1. Putting things together, we have at this point established that

$$\mathbb{E}_{\{W,z^0\}}[\|P_1 P_1^\top \widetilde{B}\|_F^2] = \|B\|_F^2 (\underline{\rho} + o_d(1)) + \frac{1}{4}\mathbb{E}_{\{W,z^0\}}[\varepsilon_{\rm rob}(f_{\rm init})]. \tag{107}$$

Similarly, noting that $P_1 P_1^\top W^\top = W^\top$ by definition of $P_1$, one has

$$\begin{aligned}
\|P_1 \widetilde{B} P_1^\top\|_F^2 &= \text{tr}(P_1 P_1^\top \widetilde{B} P_1 P_1^\top \widetilde{B}) = \text{tr}((P_1 P_1^\top B - W^\top QW)(P_1 P_1^\top B - W^\top QW)) \\
&= \text{tr}(P_1 P_1^\top B P_1 P_1^\top) - \text{tr}(P_1 P_1^\top B W^\top QW) - \text{tr}(P_1 P_1^\top W^\top WQW B) \\
&\quad + \text{tr}(W^\top QW W^\top QW).
\end{aligned} \tag{108}$$

Taking expectation w.r.t $W$ and $z^0$ then gives

$$\begin{aligned}
\mathbb{E}_{\{W,z^0\}}\|P_1^\top \widetilde{B} P_1\|_F^2 &= \mathbb{E}_W[\|P_1^\top B P_1\|_F^2] + \mathbb{E}_{\{W,z^0\}}[\text{tr}(W W^\top QW W^\top Q)] \\
&= \|B\|_F^2 (\underline{\rho}^2(1-\beta) + \underline{\rho}\beta + o_d(1)) + \frac{1}{4}\mathbb{E}_{\{W,z^0\}}[\varepsilon_{\rm rob}(f_{\rm init})].
\end{aligned} \tag{109}$$

Combining (102), (107), (109), and (28) then completes the proof of (32).

– *test error.* The proof of formula (31) build on the proof of Theorem 2 in Ghorbani et al. (2019). Let $P_2$ be a $d \times (d - \min(m, d))$ matrix such that the combined columns of $P_1$ and $P_2$ form an orthonormal basis for $\mathbb{R}^d$. Then, one computes

$$\begin{aligned}
\varepsilon_{\rm test}(f_{\rm NTL}) &:= \|f_{\rm NTL} - f_\star\|_{L^2(N(0,I_d))} = \mathbb{E}_x[|f_{\rm NTL}(x) - f_\star(x)|^2] \\
&\overset{(a)}{=} \min_{A \in \mathbb{R}^{m \times d}} 2\|\widetilde{B} - W^\top A - A^\top W\|_F^2 \\
&\overset{(b)}{=} 2\|P_2^\top \widetilde{B} P_2\|_F^2 = 2\|P_2^\top (B - W^\top QW) P_2\|_F^2 \\
&\overset{(c)}{=} 2\|P_2^\top B P_2\|_F^2 = \varepsilon_{\rm test}(f_{\rm NT}).
\end{aligned}$$

where (a) and (b) are due to arguments analogous to arguments made in the beginning of proof of Theorem 2 in Ghorbani et al. (2019) (except that our $\widetilde{B}$ plays the role of $B$ in Ghorbani et al. (2019)) and (c) is because $P_2^\top P_1 = 0 \in \mathbb{R}^{(d-\min(m,d)) \times d}$ by construction of $P_2$. Dividing through the above display by $\varepsilon_{\rm rob}(f_\star) = 4\|B\|_F^2$ then gives (31). $\square$

## I GENERAL NONLINEAR TEACHER AND STUDENT MODELS

The results we established so far are for student-teacher models which are two-layer neural networks in certain learning regimes with Gaussian data. In this section, we consider much more general scenarios, and show lower-bounds that display similar tradeoffs between test error and robustness.

Suppose the distribution $P_x$ of the features is any distribution on $\mathbb{R}^d$ which satisfies a Poincaré inequality with constant $c^2 > 0$. This means that for any smooth function $f : \mathbb{R}^d \to \mathbb{R}$, one has $\mathrm{Var}_{P_x}(f) \le c^2 \|\nabla f\|_{L^2(P_x)}^2$, where $\mathrm{Var}_{P_x}(f) := \|f - \overline{f}\|_{L^2(P_x)}^2$ is the variance of $f$ and $\overline{f} := \mathbb{E}_{P_x} f \in \mathbb{R}$ is its mean w.r.t $P_x$. For example, $N(0, \Sigma)$ verifies a Poincaré inequality with $c^2 = \|\Sigma\|_{op}$. Consider a teacher model $f_\star$ which is now **any** function in $L^2(P_x)$ with mean $\overline{f}_\star = 0$.

**Theorem I.1.** *For every smooth student model $f : \mathbb{R}^d \to \mathbb{R}$ (neural network or not!), it holds that* $\sqrt{\varepsilon_{\mathrm{test}}(f)} + c \cdot \sqrt{\varepsilon_{\mathrm{rob}}(f)} \ge \|f_\star\|_{L^2(P_x)}.$

The nature of Theorem I.1 is a tradeoff since it directly implies that the test error cannot be decreased without increasing the robustness error. In the particular case of isotropic features where $P_x = N(0, I_d)$ as considered in the preceding sections, a Poincaré inequality with constant $c^2 = 1$ is satisfied, and we deduce from the above theorem that, for any smooth student model $f$, one has

$$\sqrt{\varepsilon_{\mathrm{test}}(f)} + \sqrt{\varepsilon_{\mathrm{rob}}(f)} \ge \|f_\star\|_{L^2(P_x)}. \tag{110}$$

Of course, apart from being only one-sided, the inequality (110) is weaker than the tradeoffs established in the preceding sections, due to the square-roots in the former. However, (110) holds without any real restriction on the teacher model $f_\star$, student $f$ model, or learning algorithm / regime; it is solely a consequence of the high-dimensional geometry of the distribution of the features, manifested via the Poincaré inequality. In contrast, the tradeoffs established in the preceding sections where for student-teacher models which where two-layer neural networks in various learning regimes.

## J PROOF OF THEOREM I.1

**Theorem I.1.** *For every smooth student model $f : \mathbb{R}^d \to \mathbb{R}$ (neural network or not!), it holds that* $\sqrt{\varepsilon_{\mathrm{test}}(f)} + c \cdot \sqrt{\varepsilon_{\mathrm{rob}}(f)} \ge \|f_\star\|_{L^2(P_x)}.$

*Proof.* WLOG, assume $\varepsilon_{\mathrm{test}}(f) \le \|f_\star\|_{L^2(P_x)}$, since the claimed lower-bound trivially holds otherwise. By the Poincaré inequality, we have

$$c^2 \varepsilon_{\mathrm{rob}}(f) := c^2 \|\nabla_x f\|_{L^2(P_x)}^2 \ge \mathrm{Var}_{P_x}(f) := \|f - \overline{f}\|_{L^2(P_x)}^2, \text{ where } \overline{f} \in \mathbb{R} \text{ is mean of } f$$

$$\ge \left| \|f_\star - \overline{f}\|_{L^2(P_x)} - \|f - f_\star\|_{L^2(P_x)} \right|^2, \text{ by the triangle inequality.}$$

In particular, we have

$$c\sqrt{\varepsilon_{\mathrm{rob}}(f)} \ge \|f_\star - \overline{f}\|_{L^2(P_x)} - \|f - f_\star\|_{L^2(P_x)}$$
$$\ge \sqrt{\mathrm{Var}_{P_x}(f_\star)} - \sqrt{\varepsilon_{\mathrm{test}}(f)} = \|f_\star\|_{L^2(P_x)} - \sqrt{\varepsilon_{\mathrm{test}}(f)}.$$

where the last line follows from $\|f_\star\|_{L^2(P_x)}^2 = \mathrm{Var}_{P_x}(f_\star) \le \|f_\star - \overline{f}\|_{L^2(P_x)}^2$. The result then follows from a simple rearrangement of the terms in the above display. $\qquad\square$

## K APPROXIMATING FUNCTION VALUES DOESN'T AMOUNT TO APPROXIMATING GRANDIENTS

### K.1 DISPROVING (9)

Henceforth, for a student model $f$, consider the residue function $h := f - f_\star$. Thus, $\nabla f - \nabla f_\star = \nabla h$. In the student-teacher setup considered in our work, the student $f$ is in general mis-specified w.r.t to the teacher $f_\star$ (for example, because the students activation is arbitrary while the teacher's activation

function is fixed to quadratic), and so the residue function $h$ has no specific structure in general. Mindful of the previous remark, to disprove (1), it is sufficient to construct a subspace $H$ of the weighted Sobolev space $W^{1,2}(N(0, I_d))$ (consisting of functions $g : \mathbb{R}^d \to \mathbb{R}$ which are square-integrable w.r.t $N(0, I_d)$ with weak-derivatives which are square-integrable w.r.t $N(0, I_d)$), such that:

for every $\epsilon, C > 0$, there exists $h_0 \in H$ with $\|h_0\|_{L^2(N(0,I_d))} \leq \epsilon$ and $\|\nabla h_0\|_{L^2(N(0,I_d))} > C.$ (111)

Indeed, take $H = W^{1,2}(N(0, I_d))$, and for fixed $\alpha \in (0, 1)$, consider the sequence of residue functions $(h_n)_n$ in $H$ given by $h_n(x) := (1/n^\alpha)\sin(nx_1)$, for every positive integer $n$, and $x = (x_1, \ldots, x_d) \in \mathbb{R}^d$. Note that a constructive way for realizing such residue functions is by taking $B = 0$ in the teacher model $f_\star$, and activation function $\sigma(t) \equiv \sin(t)$ in the student model $f$. Now, a simple computation gives $\nabla h_n(x) = (1/n^\beta)\cos(nx_1)e_1$, where $\beta := 1 - \alpha \in (0, 1)$ and $e_j$ is $j$th standard basis vector in $\mathbb{R}^d$. Furthermore,

$$\|h_n\|^2_{L^2(N(0,I_d))} = n^{-2\alpha}\mathbb{E}_{x_1 \sim N(0,1)}[\sin^2(nx_1)] = n^{-2\alpha}e^{-n^2}\sinh(n^2)$$
$$= n^{-2\alpha}(1 - e^{-2n^2})/2 \xrightarrow{n \to \infty} 0,$$
$$\|\nabla h_n\|^2_{L^2(N(0,I_d))} = \mathbb{E}_{x_1 \sim N(0,1)}[n^{2\beta}\cos^2(nx_1)] = n^{2\beta}e^{-n^2}\cosh(n^2)$$
$$= n^{2\beta}(1 + e^{-2n^2})/2 \xrightarrow{n \to \infty} \infty.$$

Thus, (2) holds and we conclude that the implication (1) claimed by the reviewer fails in general. $\square$

## K.2 SOME EXCEPTIONAL CASES WHERE (1) HOLDS

Let us roundup by noting that (1) can be true in very specific circumstances. For example, if the activation function $\sigma$ of the student is quadratic, just like the teacher model $f_\star$. Indeed, in this case the set of all residue functions $f - f_\star$ is contained in the set of polynomials of degree at most 2 in $d$ real variables; this is a finite-dimensional subspace $H = P_2(\mathbb{R}^d)$ of $L^2(N(0, I_d))$. In fact, it is easy to show via a simple counting argument that $\dim(H) = \dim(P_2(\mathbb{R}^d)) = \binom{d+2}{d} = (d+2)(d+1)/2 \lesssim d^2$. Thus, the gradient operator $\nabla$ is a finite-rank, and therefore compact operator on $H$. It follows that the $H$-restricted operator norm $\|\nabla|_H\|_{op}$ defined by

$$\|\nabla|_H\|_{op} := \sup_{h \in H \setminus \{0\}} \frac{\|\nabla h\|_{L^2(N(0,I_d))}}{\|h\|_{L^2(N(0,I_d))}}. \tag{112}$$

is finite and thus the implication (1) holds in this case. This argument is valid whenever the linear span $H$ of the residue functions $f - f_\star$ is a finite-dimensional subspace of $L^2(N(0, I_d))$, for example linear models, or more generally, polynomial models of degree $\leq D$ (corresponding to the case where the activation function $\sigma$ of the student is a polynomial of degree $\leq D$), for some fixed integer $D \geq 1$; indeed $H = P_D(\mathbb{R}^d)$ in this case, and has dimension $\binom{d+D}{d} = (d+D)(d+D-1)\ldots(d+1)/D! \lesssim d^D$.

