# OpenReview forum: "On the (Non-)Robustness of Two-Layer Neural Networks in Different Learning Regimes"
_ICLR.cc/2023/Conference — Submitted to ICLR 2023_

### Official Review · Reviewer_7Nsv · 2022-10-17

**Confidence:** 4
**Correctness:** 4
**Technical Novelty And Significance:** 3
**Empirical Novelty And Significance:** Not applicable
**Recommendation:** 6

**Clarity, Quality, Novelty And Reproducibility:**

The paper is overall clear. The trade-off between test-error and robustness is original. The technical tools are close to those of Ghorbani et al., (2019).

There are some typos, e.g., "the network with $m$" (end of the first paragraph of 'Related works') and "For simplicity, simplicity of this experiment," (caption of Figure 2).

**Strength And Weaknesses:**

Strengths:

(1) The problem is well motivated and the particular measure of robustness/sensitivity is well discussed (in Appendix C).

(2) The trade-off exhibited is, to the best of my knowledge, novel.

(3) Even though the assumptions are sometimes strong (see the weakness (2)), it is always remarkable to have an exactly solvable model, leading to a closed-form expression for the robustness.


Weaknesses:

(1) From the technical standpoint, the paper is somewhat incremental, since the tools are essentially those of Ghorbani et al., (2019). If that's not the case, I would like to encourage the authors to highlight their technical innovations in the rebuttal.

(2) The assumptions on the model are rather strong, although I understand that they are taken from Ghorbani et al., (2019). In particular:

(a) Having a teacher model which is a quadratic function is rather restrictive.

(b) I am actually wondering how crucial Condition 4.3(A) actually is. If I understand correctly, it basically ensures that the activation function is centered. In Appendix F.3, the effect of centering is discussed. Given this discussion, I wonder about what happens when Condition 4.3(A) is violated. Is the model less robust (as in the case of the neural tangent approximation with an initialization)?

(3) The authors show that having an initialization term in the neural tangent approximation (and potentially not being able to fulfil Condition 4.3(A), see remark above) leads to less robustness. It is not clear to me whether this actually means a worse resistance to adversarial examples or whether this is just an effect of the particular setting taken into account (and, more specifically, of the particular metric for robustness being chosen). Can the authors comment on this point?

(4) One of the key messages of the paper seems to be that the trade-off between generalization and robustness is of the form $\tilde\varepsilon_{\rm test}+\tilde\varepsilon_{\rm rob}=1$. I have three comments about this.

(a) This same trade-off is proved for the three settings taken into account. Does this mean that training with SGD, doing random features or the neural tangent approximation makes no difference as far as robustness is concerned? This would be a bit strange.

(b) As a matter of fact, if I understand correctly the paper, $\tilde\varepsilon_{\rm test}+\tilde\varepsilon_{\rm rob}=1$ holds for random features and neural tangent only in some corner cases ($\rho\to\infty$ for random features and $\beta=1$ for neural tangent). Can the authors comment on why, in these corner cases, one would expect that the trade-off is the same in different models?

(c) Finally, can the authors comment on the trade-off when it does not have the form $\tilde\varepsilon_{\rm test}+\tilde\varepsilon_{\rm rob}=1$? Theorem 4.1 and 5.1 hold more in general and should allow to discuss a more general trade-off.

(5) How does Theorem 6.1 relate to the body of literature that uses concentration of measure to show the inevitability of adversarial examples, see e.g. Mahloujifar et al., (2018)?

**Summary Of The Paper:**

The paper studies the trade-off between robustness and test error in a setting in which (i) the teacher is a quadratic function, (ii) infinite training data are available, and (iii) a proportional scaling between the input dimension and the neural network width is assumed. The authors provide results for 3 different regimes: (i) two-layer networks trained via SGD (Theorem 3.1), (ii) random features (Theorem 4.1), and (iii) neural tangent approximation, with and without initialization (Theorem 5.1-5.2). The test error in such models was computed in previous work by Ghorbani et al. (2019) and the contribution of this paper consists in computing a certain metric of robustness/sensitivity using rather similar tools (coming from high-dimensional statistics). By doing so, the authors are able to exhibit an interesting balance between the generalization error and the robustness to perturbation. Finally, a lower bound is provided for a general non-linear teacher/student model (Theorem 6.1). This lower bound is a relatively simple consequence of the concentration of measure phenomenon.

**Summary Of The Review:**

Overall, the paper demonstrates an interesting and novel trade-off between test error and robustness. At the technical level, the analysis is a bit incremental, as it appears to resemble closely that of Ghorbani et al., (2019). My borderline (leaning positive) score is due to the several weaknesses I pointed out above, which can potentially be resolved (at least partially) after the rebuttal.

---

> ### Author Response · Authors · 2022-11-15
> **Response to Reviewer 7Nsv**
>
> We thank the reviewer for their detailed comments and question, which we now address below.
>
> ***Technical novelty.***
> We believe our work is not incremental w.r.t [Ghorbani et al., (2019)], but rather complementary. Indeed, [Ghorbani et al., (2019)] showed that in the various learning regimes (SGD, RF, NT) test error $||f-f_*||^2$ of two-layer student neural network $f$ for approximating quadratic targets (teacher) $f_*$ is analytically solvable. In our paper, we use the same toolbox as [Ghorbani et al., (2019)], namely the theory of random matrices (RMT), to show that the robustness error $||\nabla f||^2$ is analytically solvable, and comparing with the results of [Ghorbani et al., (2019)], we deduce an tradeoff between test-error and robustness error of the form
> $$
> \text{(normalized) test error } + \text{(normalized) robustness error} = 1.
> $$
> This above trade-off (which can be seen as a kind of *law of conservation of energy*) is completely nontrivial as the functions (student and teacher) involved are highly nonlinear and there is a priori no reason why the relationship as above would hold.
>
> ***Quadratic teacher model is restrictive.***
> Indeed it would be desirable to understand more general setups. The quadratic teacher model is a minimal nonlinear models which is analaytically solvable to obtain an exact tradeoff of the form "test error + robustness error = constant". Another case which is exactly solvable and exhbits the same tradeoff is the case of linear regression with gradient-descent (GD), and stochastic gradient-descent (SGD). See Appendix B.
>
> ***Removing Condition 4.3(a).***
> This is a great point. Condition 4.3(a) is a technical assumption inspired by [Ghorbani et al., (2018)]
> We have redone the calculations, and it appears this condition $\lambda_0(\sigma) = 0$ is not required. Removing it will only modify the definition of the scalars $\overline{\lambda}$ used in the definition of the matrix $A_0$ from $\overline\lambda = \mathbb E[\sigma(z)^2] - \lambda_1(\sigma)^2$ to $\overline\lambda = \mathbb E[\sigma(z)^2]-\lambda_1(\sigma)^2 - \lambda_0(\sigma)^2$.
> As for the Condition $\lambda_2(\sigma) \ne 0$, this just for sanity reasons. Without this condition, the student neural network cannot even learn the teacher (a quadratic function). This is already apparent in Lemma F.2 which we established in the appendix, but didn't fully exploit. So yes, Condition 4.3(a) can be removed.
>
> ***Does the trade-offs mean that the robustness is thesame in SGD, RF, NT, etc. ?***
> No. The correct interpretation of our results is as follows. As far as test error and robustness error are concerned, the regimes SGD, RF, NT, live on the same curve $\widetilde \varepsilon_{test}(f) + \widetilde \varepsilon_{rob}(f) = 1$. This decomposition is a tradeoff, as it implies that one error term can only decrease at the expense of the other one increasing. In particular, for the same value of hidden number of neurons, SGD and NT will achieve a small value of (normalized) test error $\varepsilon_{test}(f)$ than RF (this was the main finding in [Ghorbani et al. (2019)]), but this is only at the expense of a higher value of robustness error $\varepsilon_{rob}(f)$ which is closer to $1$, corresponding to the robustness error of the teacher model $f_*$ (as shown in our paper).
>
> Finally, in the case of NT regime, we only obtain a tradeoff of the form "test error + robustness error = 1" in the case $\beta=1$ (meaning that the covariance matrix $\Gamma$ of the hidden neurons at initialization is perfectly aligned with, i.e is proportional to the teacher coef. matrix $B$). Refer to Corollary 5.1. For general $\beta$, Theorem 5.1 gives a tradeoff of the form "test error + robustness error $\ge $ constant".
>
> ***Connection between Theorem 6.1 and other works ?***
> The works [Mahloujifar et al. (2018); ...] are concerned with classification, while our work is concerned with regression. Notwithstanding, the aforementioned works as well as ours exploit the geometric phenomenon of measure-concentration (MC) in different ways. The works are in the classification setting and they use MC is used to show that an error set $E$ (e.g the set of cats classified as dogs by a model $f$) has positive measure, then its $\delta$-vicinity (corresponding to images of cats which are either misclassified as dogs by a perturbation of size at most $\delta$) has measure which tends to  $1$ exponentially fast as a function of $\delta$. Our Theorem 6.1 is in the regression setting and we use MC concentration is used in the form of the Poincar\'e inequality to lower-bound the Dirichlet energy (i.e $\varepsilon_{\text{rob}}(f)$) of a function $f$ by its $L^2(P_x)$ norm. We exploit this lower-bound further to obtain a one-sided trade-off of the form $\sqrt{\varepsilon_{test}(f)} + c\sqrt{\varepsilon_{rob}(f)} \ge 1$, where $c^2=c^2(P_x)$ is the Poincar\'e constant of $P_x$.
>
> We hope the above clarifications address the comments and questions raised by the reviewer.

---

> > ### Author Response · Authors · 2022-12-07
> > **Did our response address your concerns ?**
> >
> > Dear reviewer,
> >
> > We would be grateful if you could tell us whether our response effectively addressed your concerns, and if there are any remaining issues.
> >
> > Thank you in advance.

---

### Official Review · Reviewer_d515 · 2022-10-19

**Confidence:** 4
**Correctness:** 2
**Technical Novelty And Significance:** 2
**Empirical Novelty And Significance:** Not applicable
**Recommendation:** 3

**Clarity, Quality, Novelty And Reproducibility:**

The presentation is clear but the interpretation of their result seems irrelevant (at least to me)

**Strength And Weaknesses:**

I do not know how to assess the results of this paper. I feel that it is **not even wrong** since the result has nothing to do with adversarial robustness. If I understand correctly, the theoretical result basically means that when $f-f^*$ is small, $||\nabla f||$ is close to $||\nabla f^*||$.  Here $f$ and $f*$ are our model and target function, respectively. In other words, the relative robustness approaches $1$ when the approximation error is decreasing toward zero.

First, this result is kind of irrelevant, since we only need the approximant and the target function to satisfy a **coercive condition**: that the smallness of $||f-f^*||$ can imply the smallness of $||\nabla f - \nabla f^*||$. This can be seen in Section 6, where the authors actually prove the trade-off for any functions that satisfy the Poincare inequality.

Second, I think the result has nothing to do with the vulnerability of ML models. As the approximation error is small, we have $||\nabla f||\approx ||\nabla f^*||$, which in fact suggests that our model is as robust as the ground truth. This is almost the best we can expect. The vulnerability issue should be the case that the learned model is much more vulnerable than the ground truth but the test error is small. This is obviously not the case studied in this work.

**Summary Of The Paper:**

This paper studies the trade-off between robustness and test error for learning a quadratic target function. The models considered include two-layer quadratic nets, random feature models, and the neural tangent one. The sample size is assumed to be infinite and thus, the test error is essentially the approximation error.

The authors show that there exist a trade-off between approximation error and robustness of approximant: $robust(f) + error(f) = 1$. The authors argue that these theoretical findings can contribute to the explanation of why there exists a  trade-off between clean accuracy and adversarial robustness, a phenomenon observed in practice.

**Summary Of The Review:**

This paper provides a precise characterization of the trade-off between robustness and approximation error for a few models. Unfortunately, I find the result is kind of irrelevant. Please correct me if I misunderstandard the result.

---

> ### Author Response · Authors · 2022-11-15
> **Response to Reviewer d515**
>
> We thank the reviewer for their comments. We would like to emphasize that the main point of critique made by the reviewer is in fact invalid.
> Indeed, the reviewer claims that one would expect the following implication to hold
> $$||f-f_*|| \approx 0 \implies ||\nabla f - \nabla f_*|| \approx 0\tag{1},$$
> where the norms are w.r.t $L^2(N(0,I_d))$.
>
> The reviewer correctly observes that an implication of the form (1) would be sufficient to establish tradeoffs between test error and robustness error, which is the main quest of our paper.
> The reviewer then proceeds to make the strong claim that our results are trivial (since they would directly follow from (1)). However, the issue with this argument is precisely that the implication (1) is false in general. The proof of our result is rather nontrivial, and exploits the fine structure of two-layer neural networks in the different learning regimes considered.
>
> **Disproving (1).**
> So, let us provide a rigorous argument for why the implication (1) claimed by the reviewer is false in general. Note that an implication of the type (1) would be reminiscent of a kind of \textbf{converse} of the Poincar\'e inequality for the standard $d-$dimensional Gaussian distribution $N(0,I_d)$, impossible in general. Henceforth, for a student model $f$, consider the residue function $h := f-f_\star$. Thus, $\nabla f - \nabla f_\star =  \nabla h$. In the student-teacher setup considered in our work, the student $f$ is in general mis-specified w.r.t to the teacher $f_\star$ (for example, because the students activation is arbitrary while the teacher's activation function is fixed to quadratic), and so the residue function $h$ has no specific structure in general. Mindful of the previous remark, to disprove (1), it is sufficient to construct a subspace $H$ of the weighted Sobolev space $W^{1,2}(N(0,I_d))$ (consisting of functions $g:\mathbb R^d \to \mathbb R$ which are square-integrable w.r.t $N(0,I_d)$ with weak-derivatives which are square-integrable w.r.t $N(0,I_d)$), such that:
> \begin{eqnarray}
> \forall \epsilon,C > 0, \exists h_0 \in H\text{ with }||h_0||\le \epsilon \land ||\nabla h_0||> C.
> \end{eqnarray}
>
> Indeed, take $H=W^{1,2}(N(0,I_d))$, and for fixed $\alpha \in (0,1)$, consider the sequence of residue functions $(h_n)$ in $H$ given by $h_n(x) := (1/n^\alpha)\sin(n x_1)$, for every positive integer $n$, and $x=(x_1,\ldots,x_d) \in \mathbb R^d$. Note that a constructive way for realizing such residue functions is by taking $B = 0$ in the teacher model $f_\star$, and activation function $\sigma(t) \equiv \sin(t)$ in the student model $f$. Now, a simple computation gives $\nabla h_n(x) = (1/n^\beta)\cos(nx_1) e_1$, where $\beta := 1 - \alpha \in (0,1)$ and $e_j$ is $j$th standard basis vector in $\mathbb R^d$. Furthermore,
> $$
>     ||h_n||^2 = n^{-2\alpha}\mathbb E_{x_1 \sim N(0,1)}[\sin^2(n x_1)] = n^{-2\alpha}e^{-n^2}\sinh(n^2) = n^{-2\alpha} (1 - e^{-2n^2})/2 \overset{n \to \infty}{\longrightarrow} 0,
> $$
> and
> $$
>     ||\nabla h_n||^2 = \mathbb E_{x_1 \sim N(0,1)}[n^{2\beta}\cos^2(n x_1)]  = n^{2\beta}e^{-n^2}\cosh(n^2)\ = n^{2\beta}(1 + e^{-2n^2})/2 \overset{n \to \infty}{\longrightarrow} \infty.
> $$
> Thus, (2) holds and we conclude that the implication (1) claimed by the reviewer fails in general. $\Box$
>
> **Some exceptional cases where (1) holds.** Let us roundup by noting that (1) can be true in very specific circumstances. For example, if the activation function $\sigma$ of the student is quadratic, just like the teacher model $f_\star$. Indeed, in this case the set of all residue functions $f-f_\star$ is contained in the set of polynomials of degree at most $2$ in $d$ real variables; this is a finite-dimensional subspace $H
> = P_2(\mathbb R^d)$ of $L^2(N(0,I_d))$. In fact, it is easy to show via a simple counting argument that $\mbox{dim}(H) = \mbox{dim}(P_2(\mathbb R^d)) = {d+2\choose d}  = (d+2)(d+1)/2 \lesssim d^2$. Thus, the gradient operator $\nabla$ is a finite-rank, and therefore compact operator on $H$. It follows that the $H$-restricted operator norm $||\nabla|_H||$ of the gradient operator restricted to $H$
> is finite and thus the implication (1) holds in this case. This argument is valid whenever the linear span $H$ of the residue functions $f-f_\star$ is a finite-dimensional subspace of $L^2(N(0,I_d))$, for example linear models, or more generally, polynomial models of degree $\le D$ (corresponding to the case where the activation function $\sigma$ of the student is a polynomial of degree $ \le D$), for some fixed integer $D \ge 1$; indeed $H = P_D(\mathbb R^d)$ in this case, and has dimension ${d + D\choose d} = (d+D)(d+D-1)\ldots(d + 1) / D! \lesssim d^D$.
>
> In the revised manuscript, we have added Remark 2.1 (and Appendix K) to highlight the above argument.

---

> > ### Comment · Reviewer_d515 · 2022-11-18
> > **It would be helpful if the authors can read my comment carefully!**
> >
> > Thank for your comment. However, I feel you misunderstood my comment. **I explicitly wrote "need the approximant and the target function to satisfy a *coercive condition*"**.   Hence, the counterexample functions  provided that do not satisfy the coercive condition seems irrelevant to my comment. Of course,  it might be not trivial to verify that some models satisfy this **coercive condition**, of couse as can been  seen from your derivations.  However, technically speaking, I do not see why it is interesting or relevant to verify the coercive condition.
> >
> > Obviously the above is not my concern but my interpretation of your result. My major concern is that the authors prove that as the model approach to ground truth, the model's robustness  approachs to the target function's robustness. Then, the author implicitly claim that **the model is **vulnerable** when the model is in fact as robust as the groundth truth**. This really does not make sense to me. I do not understand why requiring a model to be more robust than the ground truth is relevant.

---

> > > ### Author Response · Authors · 2022-11-18
> > > **Follow-up response to Reviewer d515**
> > >
> > > Thanks for your clarification. Our focus in the paper is in fact on deriving precise results for a few well-chosen learning regimes for two-layer neural networks, which is highly non-trivial (contrary to the remark in your previous post). This is the point we were making in our previous reply.
> > >
> > > When it comes to interpreting our results from the lens of "adversarial vulnerability", we agree with your point regarding the comparison with the robustness of the target model. However, we note that the assumption of a quadratic target function (teacher) is often intended as an approximation/mis-specification of the true data-generating process. This is a reasonable approximation for what can truly be captured by many models in certain high-dimensional regimes (see, e.g., [Ghorbani et al. 2021](https://arxiv.org/abs/1904.12191)). Then, having $\varepsilon_{rob}(f) = 1$ may indeed correspond to a vulnerable model, as illustrated by the following examples:
> > >
> > > (i) we might have a ground truth target function that is piecewise constant (e.g. a step function), for which the quadratic approximation (in $L^2(N(0,I))$) has much higher Dirichlet norm; in this case, $\varepsilon_{rob}(f) = 1$ may actually correspond to a model with much worse robustness than the ground truth.
> > >
> > > (ii) our quadratic model may be an approximation of an anticausal model (y causes x) for which there are natural trade-offs between accurate-but-vulnerable models, and robust-but-inaccurate models, as in [Tsipras et al.](https://arxiv.org/abs/1805.12152). By analogy to Tsipras et al, we may view the different eigenvectors of $B$ as representing robust or non-robust features depending on their eigenvalues.
> > >
> > > As per the reviewer’s assessment of the technical novelty and correctness of our work, we would greatly appreciate it if the reviewer could clearly state what is technically incorrect in our work, and whether our analysis appears in previous work.

---

> > > > ### Comment · Reviewer_d515 · 2022-11-19
> > > > **I do not find the explanation on the vulnarability of groundth truth/target function reasonable.**
> > > >
> > > > ## About the technical part
> > > > First, I looked at again my comment and do not find any assessment of the correctness of your result. I do not understand why you keep suggesting that I should clearly state "what is technically incorrect in our work".
> > > >
> > > > Second, I also did not comment on whether your analysis is technically novel or not, since this is not my point. My point is that your interpretation of the results does not make sense to me, which makes me feel that the technical part is irrelevant to understanding the vulnerability of ML models.
> > > >
> > > > The only problem is that I used the word "trivial" (I will change it), which is not accurate and appropriate. Perhaps a better word is "irrelevant". What I want to point out there is that for obtaining similar conclusions,  one only needs to verify the coercive condition to be satisfied by these models.  I think the coercive condition is much more essential than the so-called "precise trade-off" equality.  In my humble opinion, the trade-off equality is precise but does not provide insight.
> > > >
> > > > ## About the interpretation
> > > >
> > > > First, in your paper, in particular the abstract, you keep claiming that your theoretical analysis can lead understanding of adversarial robustness, and most citations are also about adversarial robustness. But I do not find the trade-off equation relevant to this topic.
> > > >
> > > > Second, I thank the authors for the extra interpretations provided above. I did not read (Tsipras et al. 2018) and I do not think it is my reponsibility to read  extra references. To be honest, I do not fully understand the argument. But the following are my thoughts.
> > > >
> > > > - If I understand correctly, you are arguing that the target function is not the ground truth. Hence, the target function can be vulnerable. But if this is the case, the error becomes meaningless for the ground truth, consequently, you cannot claim the trade-off.
> > > >
> > > > - Or are you trying to suggest that the ground truth itself is vulnerable? I do not feel this is a reasonable assumption for regression. But for classification problems, one may argue that the ground truth may not be uniquely defined, where both the robust features and non-robust features may yield the same labels. However, the analysis in your paper is not directly applicable to the classification setting, and also I do not think there exists a trade-off between approximation error and robustness in this setting. It is more often that approximating the robust features needs more expressiveness than approximsting non-robust features.

---

> > > > > ### Author Response · Authors · 2022-11-22
> > > > > **Some clarifications**
> > > > >
> > > > > Thanks for your reply.
> > > > >
> > > > > - Regarding your assessment, we were simply puzzled by your scores in the "Correctness" and "Technical novelty and significance" sections, which suggest incorrectness and lack of novelty. Perhaps this was unintentional?
> > > > >
> > > > > - Regarding the interpretation of vulnerability, it seems that we are getting closer to an agreement. Indeed, the ability to predict using either robust or non-robust features in a classification setting is also meaningful in regression. In particular, if we start with a classification setup as you are describing, with $y \in \{0,1\}$, then the Bayes classifier for the 0-1 loss relies on $\mathbb P(y=1|x)$, which is also the target function in a regression setup with squared loss, since $f^*(x) = \mathbb E[y|x] = \mathbb P(y=1|x)$ in this case. If both robust and non-robust features are predictive, then the target function $\mathbb P(y=1|x)$ typically depends on both sets of features, and may thus be vulnerable, whether it is in classification or in regression. In our quadratic setup, these different features may correspond to different eigenvectors of the matrix $B$.

---

> > > > > > ### Comment · Reviewer_d515 · 2022-11-25
> > > > > > **Thanks for the clarifications**
> > > > > >
> > > > > > Thanks for the reply. I did not intend to evaluate the "correctness" and "novelty" technically there. My choice was purely based on my general assessment of your result. For instance, I think your claims on the implication of the precise trade-off equality are incorrect. Unfortunately, there is no choice in the "Correctness" section that matches my assessment.  I apologize if it makes you feel confused. I will change the choice slightly.
> > > > > >
> > > > > > Regarding the Bayes optimal classifier, I do not think what you explained above is convincing.
> > > > > >  - First, in my understanding of classification, it is more relevant to assume both robust features and non-robust features can lead to nearly 100% accuracy. Then, the problem is that our model may only learn the non-robust features because of either the bad choice of training procedure or the model is not complex enough to represent robust features. In my opinion, this type of assumption matches better with practical observations.
> > > > > >
> > > > > > - Second, even we assume that robust and non-robust features orthogonally contribute different proportions of accuracy. It still does not fit the setup in your paper. Let us assume that robust features contribute A% accuracy and B% sensitivity, and robust features contribute the remaining  (100-A)% accuracy but (100-B)% sensitivity. Here the sensitivity is measured by e.g., the Sobolev seminorm.  Then,  if the robust features are learned well, we have the error to be (100-A)% and sensitivity B%. To make your trade-off satisfied, one needs 100-A +B=100.   Hence, A=B. In addition, since non-robust features should be much more sensitive, we should have 100-B>>B.  Without loss of generality, suppose B=5.  This leads to the conclusion that robust features only contribute A%=B%=5% accuracy. This means that robust features should contribute much less than non-robust features to the accuracy. I do not think this is a reasonable assumption that matches practice.

---

> > > > > > > ### Author Response · Authors · 2022-11-29
> > > > > > > **Some further points**
> > > > > > >
> > > > > > > We thank the reviewer for their detailed response.
> > > > > > >
> > > > > > > **Technical novelty and correctness.**
> > > > > > > We are glad you are willing to increase your evaluation (scores) of our work to better represent your current understanding, and are looking forward to this.
> > > > > > >
> > > > > > > **Interpretation of our results.**
> > > > > > > Overall, it appears there is still some confusion regarding the assessment of our work. It seems there are two possibilities:
> > > > > > >
> > > > > > > - If the reviewer is wondering whether our results predict a tradeoff between accuracy and robustness in the setup considered in our paper, the answer (for the distributional setup clearly stated in our paper), then answer is yes. As far as we can tell, we have converged on this point in the course of this rebuttal.
> > > > > > >
> > > > > > > - If the reviewer is wondering whether the setup (distributional assumptions, etc.) in our work are realistic for real-world datasets, then we believe the answer is beyond the scope of our paper. Such a question can be asked about all theoretical works on robustness (e.g Tsipras et al. (2019)) or most of learning theory in general. The point of our work is to understand the compromises between robustness and accuracy induced by structure of the ground-truth and the choice of training procedure (i.e the learning regime).
> > > > > > >
> > > > > > > The setup considered in our work is simple enough to be analytically solvable, but complex enough to exhibit nontrivial phenomena (tradeoff between accuracy and robustness, role of over-parametrization, role of initialization, universality w.r.t choice of student activation function, etc.).
> > > > > > > We don't claim that this setup is representative of the complexities present in real-world datasets. We can rephrase the abstract of our paper to reflect this more clearly.
> > > > > > >
> > > > > > > **Would robust features contribute (much) less than non-robust features to the overall accuracy ?**
> > > > > > > The reviewer brings up a great point by observing that reconciling our tradeoffs with real-world datasets might lead to uneasy constraint that: *robust features contribute much less than non-robust features, to the overall accuracy*. We argue that
> > > > > > >
> > > > > > > - this constraint is not as unnatural as it might seem (especially if we remove the qualification "much"), and
> > > > > > >
> > > > > > > - even if this constraint is judged problematic, its already present in foundational works like Tsipras et al. (2019), where the Bayes-optimal model ends up relying exclusively on non-robust features.
> > > > > > >
> > > > > > > Let us make the above argument rigorous. For clarity of exposition, let the teacher / ground-truth coefficient matrix $B$ be a diagonal matrix with diagonal entries $\lambda_1 \ge \lambda_2 \ge \ldots \ge \lambda_d \ge 0$. For any $k \in [d]$, let $f_\star^{\le k}$ be a version of $f_\star$ which only uses the top $k$ features, i.e $f_\star^{\le k}(x) :=  \sum_{j=1}^{k}\lambda_j x_j^2 + b_0$, where $b_0 := -\mbox{tr}(B) = -\sum_{j=1}^d \lambda_j$. For example, such a model would arise by running SGD on a student neural network with $k$ hidden neurons and quadratic activation function, as shown in Theorem 3 of Ghorbani et al. (2019).
> > > > > > > In the Bayes-optimal model $f_\star$, the marginal contribution of the first $k$ features to the accuracy is $A_k\%$, where $A_k=100a_k$, with $a_k \in [0,1]$ given by
> > > > > > > $$
> > > > > > > a_k\cdot||f_\star||^2 := ||f_\star||^2 - ||f_\star-f_\star^{\le k}||^2 = 2\sum_{j=1}^d \lambda_j^2 - 2\sum_{j=k+1}^d \lambda_j^2  = 2\sum_{j=1}^{k}\lambda_j^2.
> > > > > > > $$
> > > > > > > Now, as the reviewer correctly observes, combining our established tradeoffs test error and robustness / sensitivity (as measured via Dirichlet energy), one of the following practically-sound conditions
> > > > > > >
> > > > > > > - (i) the "robust features" contribute more (resp. less) than "non-robust features" to the overall accuracy (resp. sensitivity), or
> > > > > > >
> > > > > > > - (ii) the "robust features" contribute much more (resp. much less) than "non-robust features" to the overall accuracy (resp. sensitivity),
> > > > > > >
> > > > > > > (strictly speaking, the reviewer only considers condition (ii)) would yield the following *ontological* constraints: if there are $k$ "robust features", then necessarily, $a_k \le 1 - a_k$ under condition (i) and $a_k \ll 1-a_k$ under condition (ii), that is
> > > > > > >  - (a) $\sum_{j=1}^k \lambda_j^2 \le \sum_{j=k+1}^d \lambda_j^2$, for compatibility with (i), OR
> > > > > > > -  (b) $\sum_{j=1}^k \lambda_j^2 \ll \sum_{j=k+1}^d \lambda_j^2$, for compatibility with (ii).
> > > > > > >
> > > > > > > The first constraint is of course much weaker than the second (originally mentioned by the reviewer). Now, observe that since the spectral parameters $\lambda_1,\lambda_2,\ldots$ and the cutoff index $k$ are free parameters, the above constraints (especially the first) are not very restrictive, and would be compatible with the setup of our paper.
> > > > > > >
> > > > > > > Finally, constraint (a) relates to the fact that as the learner strives to fit the ground-truth better, it is forced to pick up non-robust features too. Our work underlines the fact that in order to pick up more robust than non-robust features, one should consider other algorithms (regularized models, adversarial training, etc.).

---

### Official Review · Reviewer_qTks · 2022-10-22

**Confidence:** 4
**Correctness:** 4
**Technical Novelty And Significance:** 3
**Empirical Novelty And Significance:** Not applicable
**Recommendation:** 8

**Clarity, Quality, Novelty And Reproducibility:**

Appart from the above mentioned issues, I have the following questions:

1. There exists different definitions on over-parameterization, e.g., $m/d \gg 1$ in (Allen-Zhu, 2018, https://arxiv.org/abs/1811.03962) that this paper used or $m/n \gg 1$ in (Belkin et al. 2019). I suggest the authors clarify this.

2. This work shows that over-parameterization hurts robustness, which is contrary to (Bubeck and Sellke, 2021) as well as other results on robustness. More discussion is needed.


3. This work uses the gradient norm in a L2-integrable space in the sense of expectation as a robustness metric, called Dirichlet energy. In fact, this average-case robustness view also exists in [1] under different initializations as well as the over-parameterization. (authors are excused for this paper because this paper was recently posted). Besides, I’m wondering that, the tradeoff between robustness and generalization in this paper can be extended to other robustness metrics, e.g., [1] and Lipschitz constant?

[1]  Zhu, Z., Liu, F., Chrysos, G.G. and Cevher, V. Robustness in deep learning: The good (width), the bad (depth), and the ugly (initialization). NeurIPS 2022.

4. It is unclear to me why the input dimension $d$ is used in the assumption $d || \Gamma ||_{op} = O(1)$.
Normally $ || \Gamma ||_{op} = O(1)$ makes sense in RMT.

5. It is unclear to me why the neural networks in Section 3 exhibit a form of feature learning?

**Minor issues:**

The studied test error in this paper works in the approximation theory view, but I suggest the authors adopt the estimation/test error instead of approximation error, leading to extra confusion to readers.


**Strength And Weaknesses:**

**Pros:**

1. In two-layer neural networks with polynomial activation functions trained by SGD, random features, the neural tangent kernel for two-layer networks, the tradeoff between the generalization (excess risk) and the robustness (gradient norm) is given.
2.  a lower bound between the test error and the robustness over general data is given.

**Cons:**

I’m familiar with this topic equipped with high dimensional statistics, and like this theoretical result on the trade-off between generalization and robustness. Nevertheless, I don’t like the writing style: mixing with Ghorbani et al. (2019) throughout this paper, e.g., problem settings, generalization results, random features regime and neural tangent kernel regime.
For example, Section 4.2 has already been studied in Ghorbani et al. (2019) or can be easily obtained (e.g., Eq. 20).

I strongly suggest the authors rewrite this paper to emphasize their main contribution on robustness. Besides, a table is needed to summarize the problem settings and results presented in Section 3, 4, 5.


**Summary Of The Paper:**

This paper analyzes generalization and robustness properties (measured by gradient norm) of two-layer neural networks in different learning regimes under the high dimensional setting with Gaussian data. The tradeoff between them is theoretically given for such two-layer neural networks trained by SGD, the random features regime, and the NTK regime. Besides, a lower bound between the test error and the robustness over general data is given, which implies that the test error cannot be decreased without increasing the robustness error.


**Summary Of The Review:**

This paper give a rigious theoretical evidence on the trade-off between robustness and generalization in high dimensional settings under two-layer neural networks. I vote for acceptance of this paper but strongly suggest the authors re-organize this work for better presentation.

---

> ### Author Response · Authors · 2022-11-15
> **Response to Reviewer qTks**
>
> We thank for reviewer for the insightful remarks and great questions.
>
> ***Definitions of over-parametrization and constrast with BLN21.***
> Note that the term "over-parametrization" is not used in our paper in the same sense as in [Bubeck et Sellke (2021)], or BLN21 for short. In BLN21, the setup is finite samples, and over-parametrization means $m$ is substantially larger than $n/d$, where $n$ is the sample size, $d$ is the input-dimension, and $m$ is the network with (i.e number of neurons in the hidden layer). In our work, we focus on the infinite-sample case ($n=\infty$), and over-parametrization means $m / d$ is large. Thus the claims in our paper and BLN21 are not directly comparable.
>
> ***Does our work show that over-parametrization hurts robustness ?***
> This is a great question. In (Ghorbani et al. 2019),  it was shown that for regression with two-layer neural networks in certain learning regimes like SGD, NT, or RF (with the right covariance matrix $\Gamma \propto B$), over-parametrization of the student network is sufficient for obtaining zero test error. What our paper shows is that in this over-parametrized regimes, the robustness error of such a student model also approaches that of the teacher model, i.e there is a tradeoff between these two types of errors.
>
> Note that BLN21  does not argue that over-parametrization is needed for robustness. It argues that any two-layer neural network (with Lipschitz activation function, ...) which \emph{interpolates} training data needs over-parametrization if it is to be robust. In fact, it has been shown that over-parametrization can be detrimental to robustness.
> Indeed, in the case of linear models, see
> -- [Javanmard et al. (2020)] *Precise tradeoffs in adversarial training for linear regression* \url{http://proceedings.mlr.press/v125/javanmard20a/javanmard20a.pdf}.
>
> -- [Donhauser etl al. (2021)] *Interpolation can hurt robust generalization even when there is no noise* \url{https://proceedings.neurips.cc/paper/2021/file/c4f2c88e16a579900657c18726641c81-Paper.pdf}.
>
> For the non-linear case, see [Hassani \& Javanmard (2022)] *The curse of overparametrization in adversarial training: Precise analysis of robust generalization for random features regression* \url{https://arxiv.org/abs/2201.05149} showed that over-parametrization is detrimental for adversarial training, while [Zhu et al. (2022)] *Robustness in deep learning: The good (width), the bad (depth), and the ugly (initialization)* \url{https://arxiv.org/pdf/2209.07263.pdf} show that as the width $m$ of a neural network is increased, there is transition from over-parametrization being detrimental, to being beneficial for robustness. More precisely,  they derived upper-bounds for robustness error which show that there critical value $m_0$ such that the robustness error is an increasing function of width $m$ in the interval $[1,m_0]$ (over-parametrization hurts robustness) and a decreasing function of $m$ in the interval $[m_0,\infty)$ (i.e over-parametrization is beneficial). This nuances the apparently contradictory findings of BLN2021 --namely, that over-parametrization is beneficial for robustness, and [Hassani \& Javanmard (2022)] --namely, that over-parametrization is detrimental to robustness.
>
> Overall, the exact role of over-parametrization in robustness remains partly unclear, even though progress is being made on the subject.
> We have enriched our manuscript with the above discussion (highlighted in magenta) to clarify the situation.
>
> ***Extension to  Lipschitz constant ?***
> Indeed, note that Sobolev-seminorm is a lower-bound for Lipschitz constant. Indeed, $\varepsilon_{\text{rob}}(f) := (\mathbb E_{P_x} ||\nabla f(x)||^2)^{1/2} \ge \sup_x ||\nabla f(x)||^2 \ge (\sup_x ||\nabla f(x)||)^2 = Lip(f)^2$. Thus any lower-bound on $\varepsilon_{\text{rob}}(f)$ (inherent in the tradeoffs we show in our paper) immediately leads to a lower-bound on Lipschitz constant.
>
> ***On the condition $d|| \Gamma||_{op} = O(1)$.*** This condition is to ensure that $\Gamma$ has trace of order $O(1)$,
>  that is $||w||^2$ is of order $O(1)$ for $w \sim N(0,\Gamma)$. This assumption is consistent with the practice of neural networks where $\Gamma$ is usually taken to be $(2/d)I_d$. Of course, the condition can be rewritten as $||\widetilde \Gamma||_{op} = O(1)$ with $\widetilde\Gamma := \Gamma/d$; in the standard practice of neural networks, $\widetilde \Gamma = I_d$.
>
>   ***On feature learning in SGD.***
>   By feature learning, we are referring to the fact that the first layer weights are learning specific directions by approximating the matrix $B$. In particular, this involves non-trivial feature selection via non-linear learning, while the other regimes (RF and NT) are linear estimators on top of non-linear but fixed features.

---

> > ### Comment · Reviewer_qTks · 2022-11-16
> > **question on SGD and Sobolev-seminorm as a lower bound for Lipschitz constant**
> >
> > Thanks for the authors' feedback on the relationship between over-parameterization and robustness.
> > There are two issues I still concern:
> >
> > - SGD: The authors mentioned that SGD is a kind of feature learning different from RF, NTK. To me, it's clear that RF, NTK belong to lazy training or "linear" regime, but it's unclear to me on SGD in the feature learning setting. Normally, these lazy regime depends on certain initialization and structure.
> >
> > - Sobolev-seminorm as a lower bound for Lipschitz constant: about the first inequality, I think the direction is incorrect. By Holder inequality, we have
> >
> > $E_{P_x} || \nabla f(x) ||^2 \leq \max_{x}  || \nabla f(x) ||^2$.
> >
> > A conterexample, if we choose f(x) = $1/3 x^3$ and $P_x \sim N(0,1)$, we have $E_{P_x} || \nabla f(x) ||^2 = 1$ but $ \max_{x}  || \nabla f(x) ||^2$ can be unbounded.
> >
> > Besides, for the second inequality, $\sup_x || \nabla f(x) ||^2 = (\sup_x || \nabla f(x) || )^2$ since $|| \nabla f(x) ||$ is non-negative.

---

> > > ### Author Response · Authors · 2022-11-16
> > > **Response to: question on SGD and Sobolev-seminorm as a lower bound for Lipschitz constant**
> > >
> > > We thank the review for the further remarks.
> > >
> > > ***Regarding the first point (SGD).*** We are not saying that SGD in general performs feature learning. Rather, in this particular setting, where we only optimize first-layer weights on an online setting, it is possible to show that the weights do adjust to match the target weights (i.e. those that correspond to rank-$m$ approximation of $B$, where $m$ is the width of the student network). For more details on this, see Section 2.3 in the Ghorbani et al. paper.
> > >
> > > ***Concerning the second point (on Sobolev seminorm).*** Indeed, as correctly noticed by the reviewer, there is a typo in the direction of the inequalities we wrote in our response. What we intended to write down is the following:
> > >
> > > $\varepsilon_{\text{rob}}(f) := \mathbb E_{P_x} ||\nabla f(x)||^2  \le \sup_x ||\nabla f(x)||^2 \le (\sup_x ||\nabla f(x)||)^2 = Lip(f)^2$.
> > >
> > > That is, robustness error is always a lower-bound for (the square of) Lipschitz constant.

---

> > > > ### Author Response · Authors · 2022-12-07
> > > > **Did our response address your concern (about SGD) ?**
> > > >
> > > > Dear reviewer,
> > > >
> > > > We were wondering whether our response addresses your last concern (about SGD), and if there are any remaining issues.
> > > >
> > > > Thank you in advance.

---

### Official Review · Reviewer_vJwo · 2022-10-24

**Confidence:** 3
**Correctness:** 4
**Technical Novelty And Significance:** 3
**Empirical Novelty And Significance:** 3
**Recommendation:** 6

**Clarity, Quality, Novelty And Reproducibility:**


- **Clarity**: the paper is written very clearly, and very technically precise. It is also relatively dense--see my suggestion below.
- **Quality**: the paper rigorously considers various settings and has interesting insights that I learned
- **Novelty**: to my knowledge, considering this setup for the robustness-generalization trade-off is novel, and moreover, the connections with initializations are also novel
- **Reproducibility** : I did not see an appendix on implementation details, please clarify in your response if the code will be made public.



### Minor points / Suggestions
- Maybe this has been also done by previous works, but I personally find it confusing that the terms *teacher-student* are used here given that teacher-student is a different problem, whereas here the teacher is not a neural network.
- In the abstract, it is not clear what is referred to by *lazy training*; either add a citation or describe it with a few words therein
- While reading the paper, I was often wondering why is this relevant -- it would be very helpful to improve the flow of the reading: e.g. prepare the reader for what follows, why the answer that follows is relevant etc. I understand that there is a lack of space and it is a strict advantage of the paper that there are many contributions, but I would advise moving some less-important details in the appendix if needed (while summarizing only in an informal theorem for e.g.) in order to improve the reading flow.


**Strength And Weaknesses:**

This paper studies the robustness of two-layer neural networks in different learning regimes and proves that there is a compromise between robustness and test accuracy.

Pros:
- the paper is well structured and, (albeit dense) it is relatively clear
- to my knowledge, understanding the tradeoff on this particular setup is novel and well motivated
- it is particularly nice that the considered setup is simple so that it can be studied thoroughly.

Cons:
- while I agree with the soundness of the robustness measure used, and I appreciate the additional details in the appendix, I am not sure to which extent the results of this paper (in the regression setting) can be generalized to the more standard classification setting.
- there are missing comparisons with existing works (see below)

## Missing comparisons with existing works
- The work of Zhang et al. (TRADES) which focuses on the classification setting is also discussing the tradeoff between robustness and accuracy
- Specifically for the regression case,  [Rothenhäusler et al., 2018.] rigorously show the trade-off between robustness and test accuracy in a linear setting; it is necessary to discuss the differences.

[Zhang et al, 2019] *Theoretically Principled Trade-off between Robustness and Accuracy* -- https://arxiv.org/abs/1901.08573
[Rothenhäusler et al., 2018.] *Anchor regression: heterogeneous data meets causality* -- https://arxiv.org/abs/1801.06229

**Summary Of The Paper:**

This paper studies the trade-off between the test accuracy and robustness of two-layer neural networks, specifically concerning regression. More precisely, they focus on using a 2-layer network to learn a quadratic form.
Theoretically, it is shown that there is a clear trade-off between robustness and test accuracy for some variants of the above setup where the initialization is taken into account, as well as the neural tangent regime.
The simulations verify the findings for this specific setup.

**Summary Of The Review:**

The compromise between test accuracy and robustness has been studied before in the regression and classification setting, but this work sheds light on more detailed discussions that include initialization and other regimes, specifically for regression given quadratic form.
While broadly the question is relevant, it is not clear if these findings can be extended to the more relevant classification setting.

---

> ### Author Response · Authors · 2022-11-15
> **Response to Reviewer vJwo**
>
> We thank for reviewer for the insightful remarks and great questions.
>
> ***Other related works.*** We thank the reviewer for bringing the work [Rothenhäusler et al., 2018.] *Anchor regression: heterogeneous data meets causality* \url{https://arxiv.org/abs/1801.06229}
>  to our attention. The referenced paper studies tradeoffs between test error and distributional robustness in linear regression under distributional shifts on the marginal distribution of the covariates. Though very interesting, we believe the setup of [Rothenhäusler et al., 2018] is far from the the concern of our paper, namely tradeoffs between predictive and adversarial robustness performance in two-layer neural networks.
>  Nonetheless, we have included the mentioned paper in the paper, along with the other related works on robustness.
>
>
> The work [Zhang et al, 2019] *Theoretically Principled Trade-off between Robustness and Accuracy* \url{https://arxiv.org/abs/1901.08573} mentioned by the reviewer studies the case of classification and establishes a decomposition of robustness error as a sum of a a natural error $\varepsilon_{\text{nat}}(f)$ and error due to points close to the decision boundary. The second error is called boundary error $\varepsilon_{\text{bdy}}(f)$, and plays a role similar to the role played by our $\varepsilon_{\text{rob}}(f)$ in the context of regression. The authors then use this decomposition to derive a trade-off between natural error and boundary error. This paper is indeed relevant to the discussion and we have included it in the
> "Related works" part of our manuscript (Section 1).
>
> ***The teacher is indeed two-layer neural network.*** Indeed, the teacher model $f_\star(x) \equiv x^\top B x + b_0$ is a two-layer neural network as mentioned in the beginning of Section 2.2 of the paper. To see this, let $B=W_\star^\top W_\star$ be the Cholesky factorization of $B$ with $W_\star  \in \mathbb R^{m_\star \times d}$. Then, we can write $f_\star(x) \equiv \sum_{j=1}^{m_\star} (x^\top w_\star^j)^2 + b_0$, which is a two-layer neural network with quadratic activation function, $m_\star$ hidden neurons, hidden weights matrix $W_\star$, and output weights vector $v_\star=1_{m_\star} := (1,\ldots,1)$.
>
> ***Definition of "lazy training".***
> We use this term in the sense of [Chizat et al. (2019)] *On lazy training in differentiable programming* (cited in page 1 of our manuscript), i.e the parameters of the student neural network evolve like  a linearization around the initialization.
>
> ***Reproducibility.*** The contains accompanying .zip (supplemental) well-documented python files for reproducing the experiments. The experimental details are in the figure captions. The code will be made public after publication of the manuscript.

---

> > ### Author Response · Authors · 2022-12-07
> > **Did our response address your concerns ?**
> >
> > Dear reviewer,
> >
> > We would be grateful if you could tell us whether our response effectively addressed your concerns, and if there are any remaining issues. Thank you in advance.

---

### Author Response · Authors · 2022-11-15
**General Response (to all the Reviewers)**

We thank for the reviewers for their effort and insightful comments. We posted our responses individually to each reviewer, addressing their questions and concerns. We re-uploaded the paper with a few changes (colored in magenta) based on the reviewers’ suggestions.

**Recap of our main contributions (for reference):**
Our work considers predicting a nonlinear  output using (with unknown coefficient matrix $B$) two-layer neural networks. The setup considered in our work is simple enough to be analytically solvable, but complex enough to exhibit nontrivial phenomena (tradeoff between accuracy and robustness, role of over-parametrization, role of initialization, universality w.r.t choice of student activation function, etc.).

- We establish systematic tradeoffs between test error (as measured by prediction quality of test data) and robustness error (as measured by average norm of gradients out predicted outputs w.r.t inputs at test time) in different learning regimes: SGD, random features (RF), neural tangent (NT), etc. These tradeoffs are of exact form: "test error + robustness error = 1".

- These tradeoffs mean that robustness error can only decrease at the expense of increasing test error and vise versa.

- Our theoretical results are confirmed by experiments. We observe the predicted tradeoffs between robustness error and test error

- Motivated by our work, we postulate that such tradeoffs are a combined inductive bias of the certain model classes and learning regimes, and merits future investigation. In particular, it would be interesting to characterize precisely under what settings such tradeoffs hold, or at least enlarge the scope of model classes / learning regimes (beyond linear and two-layer neural networks in SGD, RF, NT, etc. learning regimes) in which they hold. This direction was initiated in Section 6 (now moved to Appendix I, due to space constraints) of our manuscript, where established a weaker one-sided trade-off (i.e an inequality instead of an equality) between test error and robustness error.

- Our work can be seen as a first step towards a rigorous theoretical understanding of the robustness of trained neural networks, an important subject which is still understudied.

---

### Decision · Program_Chairs · 2023-01-20

**Decision:**

Reject

**Justification For Why Not Higher Score:**

I provided a detailed discussion above.

**Justification For Why Not Lower Score:**

N/A

**Metareview: Summary, Strengths And Weaknesses:**

The paper studies the robustness of a teacher-student architecture with two layers. While prior work relied on the Lipschitz constant (i.e. worst-case measure) to measure the robustness of neural networks, this paper instead studies an average measure of the gradient. This is a novel angle that the reviewers appreciated.

Initially, two reviewers were mildly positive, one was positive, and one was negative (reviewer d515). After the discussion period, reviewer d515 maintained his low score. We therefore had a Zoom meeting between the reviewers followed by a discussion with the meta-AC.

Overall, the paper does have an interesting aspect to it as it studies a novel measure of robustness (which is justified in the appendix) and provides an analytical formula for this robustness measure. From a technical point of view, the paper heavily builds on the prior results by Ghorbani et al. (2019) (for instance using their linearization results for the random matrix that appears in the robustness formula for the random feature model). This opinion is shared among all reviewers, including Reviewer qTks who is the most positive about the paper. I think the strong connection to Ghorbani et al. (2019) should be more clearly highlighted in the paper (the current writing simply says the paper is "inspired" by their paper, while many lemmas actually directly come from their paper). I do believe this is an interesting contribution that should be published at some point, but I largely agree with reviewer d515 that the paper shouldn't be published in its current form. I elaborate below.

As highlighted by Reviewer d515, the quadratic student-teacher setting is rather artificial and does not necessarily reflect what happens to practical neural networks. The major issue raised during the Zoom meeting is that the paper presents itself as an analysis of neural networks and does not sufficiently acknowledge the somewhat "artificial" setting studied in the paper (this can clearly be seen in the choice of the title, the introduction and the discussion section). This main point was, in my view, not adequately addressed by the authors during the discussion period, leaving it uncertain that the authors would address this point if the paper were to be accepted.
For instance, looking at the changes made to the paper during the rebuttal period, I do not see a serious attempt made by the authors to address this problem.

I will therefore reject this paper for now and I strongly recommend the authors rewrite the paper to address this concern. I hope this decision will not discourage the authors, I'm convinced a revised version of this paper would be worth publishing in a good venue.



**Summary Of Ac-Reviewer Meeting:**

See above.